# p97 regulates GluA1 homomeric AMPA receptor formation and plasma membrane expression

Yuan Ge [1,2,7], Meng Tian[1,7], Lidong Liu[1], Tak Pan Wong [1,5], Bo Gong [1], Dongchuan Wu [1,3], Taesup Cho[1,6], Shujun Lin[4], Jürgen Kast[4], Jie Lu[1] & Yu Tian Wang [1,3]

The α-amino-3-hydroxy-5-methylisoxazole-4-propionic acid subtype glutamate receptors (AMPARs) mediate the fast excitatory synaptic transmission in the mammalian brain and are important for synaptic plasticity. In particular, the rapid insertion of the GluA1 homomeric (GluA1-homo) AMPARs into the postsynaptic membrane is considered to be critical in the expression of hippocampal CA1 long-term potentiation (LTP), which is important for certain forms of learning and memory. However, how the formation and trafficking of GluA1-homo AMPARs are regulated remains poorly understood. Here, we report that p97 specifically interacts with and promotes the formation of GluA1-homo AMPARs. The association with p97 retains GluA1-homo AMPARs in the intracellular compartment under basal conditions, and its dissociation allows GluA1-homo AMPARs to be rapidly inserted into the postsynaptic membrane shortly after LTP induction. Thus, our results shed lights into the molecular mechanisms by which p97 regulates GluA1-homo AMPARs formation and trafficking, thereby playing a critical role in mediating synaptic plasticity.

[1] Djavad Mowafaghian Centre for Brain Health and Department of Medicine, University of British Columbia, Vancouver, BC, Canada V6T 2B5. [2] Djavad Mowafaghian Centre for Brain Health and Department of Psychiatry, University of British Columbia, Vancouver, BC, Canada V6T 2B5. [3] Translational Medicine Research Center, China Medical University Hospital, and Graduate Institutes of Immunology and Biomedical Sciences, China Medical University, Taichung, Taiwan. [4] Biomedical Research Centre, University of British Columbia, Vancouver, BC, Canada V6T 1Z3. [5] Present address: Douglas Mental Health University Institute and Department of Psychiatry, McGill University, Montreal, QC, Canada H4H 1R3. [6] Present address: Neurorive Inc., Seoul, Republic of Korea. [7] These authors contributed equally: Yuan Ge, Meng Tian. Correspondence and requests for materials should be addressed to Y.G. (email: geyuan@mail.ubc.ca) or to J.L. (email: jie.lu@mail.ubc.ca) or to Y.T.W. (email: ytwang@brain.ubc.ca)

The α-amino-3-hydroxyl-5-methyl-4-isoxazolepropionic acid receptor (AMPARs) is the primary ionotropic glutamate receptor subtype that mediates fast synaptic transmission at the vast majority of excitatory synapses in the mammalian brain. In addition, it also plays critical roles in mediating various forms of synaptic plasticity under both physiological and pathological conditions[1,2]. Structurally, AMPARs are tetrameric complexes assembled by combining four homologous subunits: GluA1–GluA4. The subunit compositions of AMPARs vary depending on brain areas as well as developmental stages. The compositions influence channel properties, trafficking and subcellular localizations[1,3–5].

In many brain areas, particularly in the principal neurons of cortex and hippocampus, the native AMPARs are predominantly GluA2-containing that are composed of the GluA1/2 or GluA2/3 subunits, with only a small subpopulation of the GluA2-lacking (likely GluA1-homo) AMPARs[6,7]. The presence of the GluA2 subunits in these heteromeric AMPARs renders their channels impermeable to calcium and a linear current–voltage relationship. In contrast, GluA1-homo AMPARs have higher calcium permeability, larger channel conductance, and an inwardly rectifying I–V relationship owing to the lack of the GluA2 subunit[3–5]. Evidence accumulated from recent studies suggests that this small subpopulation of GluA1-homo AMPARs is present in the intracellular reserve pools of AMPARs in the hippocampal CA1 neurons under basal conditions and can be rapidly translocated into synapses under certain physiological and pathological conditions. GluA1-homo AMPARs play critical roles in mediating various forms of synaptic plasticity, particularly those in the hippocampus, including long-term potentiation (LTP[8]; but also see ref. [9]), long-term depression (LTD)[10], and homeostatic synaptic scaling[11,12]. In addition, a rapid increase of GluA1-homo AMPARs has also been attributed to CA1 neuronal death following global ischemic insults[13]. Despite their physiologically and pathologically significant roles, the mechanisms by which GluA1-homo AMPARs are formed, retained intracellularly under basal conditions, and translocated into synapses during the expression of these various forms of synaptic plasticity remain largely unknown.

Using co-immunoprecipitation (Co-IP) combined with mass spectrometric analysis, we identified p97, a type II AAA ATPase also called valonsin-containing protein, as a GluA1 subunit-specific interacting protein. Our results demonstrate that p97 only interacts with the GluA1-homo AMPARs, but not with the GluA1/GluA2 heteromeric AMPARs, in hippocampal neurons. Through its specific interaction with the GluA1 subunit, p97 promotes the formation of GluA1-homo AMPARs and retains them intracellularly. Importantly, we found that following the induction of LTP, p97 rapidly dissociates from GluA1, resulting in a rapid insertion of GluA1-homo AMPARs into the postsynaptic membrane at hippocampal CA1 neurons and LTP expression.

## Results

**p97 is a GluA1-homo AMPARs interacting protein**. In order to investigate AMPARs subunit-specific interacting proteins, we raised polyclonal antibodies against the C-terminal of the GluA1 or GluA2 subunit, the two major subunits of the native AMPARs expressed in the hippocampus. The specificity of the antibodies was tested by immunoprecipitation and blotting for HA-GluA1 or GluA2 transiently expressed in COS7 cells. These two antibodies showed very high levels of selectivity without obvious cross reaction (Fig. 1a). Thus, we used these antibodies to immunoprecipitate the AMPAR complexes from the rat hippocampal homogenates. A clear band with molecular weight of about 97

kDa was found in the anti-GluA1, but not anti-GluA2, precipitates (Fig. 1b). Mass spectrometric analysis identified p97 as the putative candidate protein with the highest probability based on Mowse Score (240) and peptide sequence coverage of 54% (Table 1).

To validate these findings, we performed Co-IP using hippocampal homogenates with anti-GluA1 or anti-GluA2 antibody and immunoblotted with anti-p97 antibody. p97 could only be detected in anti-GluA1, but not anti-GluA2, co-immunoprecipitates (Fig. 1c). Thus, our Co-IP and mass spectrometry results suggest that p97 might be constitutively and specifically associated with the GluA1, but not GluA2, subunits of native AMPARs in the hippocampus. However, these results are quite surprising when considering that the majorities of native AMPARs expressed in the hippocampus are the GluA1/GluA2 or GluA2/GluA3 heteromeric receptors, with only a small population of GluA1-homo AMPARs[6]. The inability of anti-GluA2 antibodies to co-immunoprecipitate p97 in our results strongly suggests that p97 only specifically interacts with the GluA1 subunit of the homomeric, but not the GluA1 or GluA2 of the GluA1/GluA2 heteromeric AMPARs. We further tested this conjecture using COS7 cells co-transfected with p97 and HA-GluA1 or GluA2, and then co-immunoprecipitated with our anti-GluA1 and GluA2 antibodies (Fig. 1d), or commercially available anti-HA antibody (Fig. 1e). We also tested this in COS7 cells transfected with p97-GFP and HA-GluA1 or GluA2, and co-immunoprecipitated with GFP-Trap (Fig. 1f). Consistent with the specific interaction between p97 and GluA1, we found that p97 could only be co-immunoprecipitated with the GluA1, but not GluA2 subunits (Fig. 1d–f).

AMPAR subunit contains a large extracellular N-terminal domain, three trans-membrane spanning domains (TM1, TM3, and TM4), a re-entry hairpin loop (M2), an extracellular loop (S2) between TM3 and TM4, and an intracellular C-terminal domain (Fig. 1g). To investigate which domain of GluA1 is involved in the interaction with p97, we designed several deletion or swap versions of GluA1 (Fig. 1g) and individually cotransfected them with p97. The Co-IP identified that p97–GluA1 interaction was not affected by the deletions of TM1, M2, TM3, S2, TM4, or C-terminal of GluA1; but it was totally abolished by swapping GluA1 N-terminal with GluA2 N-terminal (A2N/A1) (Fig. 1h), suggesting the interaction domain is in the N-terminal of GluA1. To further confirm this and test whether the interaction is a direct binding, GST pull down assay was performed using purified GST-fusion protein of GluA1 N-terminal (GST-GluA1N) or GST-fusion protein of GluA2 N-terminal (GST-GluA2N) and 6×his-tagged p97 (Fig. 1i). The result shows that the purified recombinant p97 can only be specifically pulled down by the GST-GluA1NT, but not GST-GluA2NT, demonstrating that p97 is specifically associated with GluA1 via a direct interaction involving the N-terminal domain of GluA1.

**p97 promotes the formation of GluA1-homo AMPARs**. As aforementioned, most of the native GluA1 forms the GluA1/GluA2 heteromeric AMPARs, and only a small portion of them self-dimerizes to form the homomeric AMPARs in the hippocampus[6]. It is critically important to know if and how p97, through its specific interaction with the GluA1 subunit, affects GluA1's ability to form the homomeric or heteromeric AMPARs. To this end, we first co-transfected equal amounts of p97 and HA-GluA1 plasmids with various amount of nontagged GluA2 plasmids in COS7 cells. As shown in Fig. 1j, with the increased expression of GluA2, the ability of anti-GluA1 to co-immunoprecipitate p97 decreased, suggesting a dose-dependent

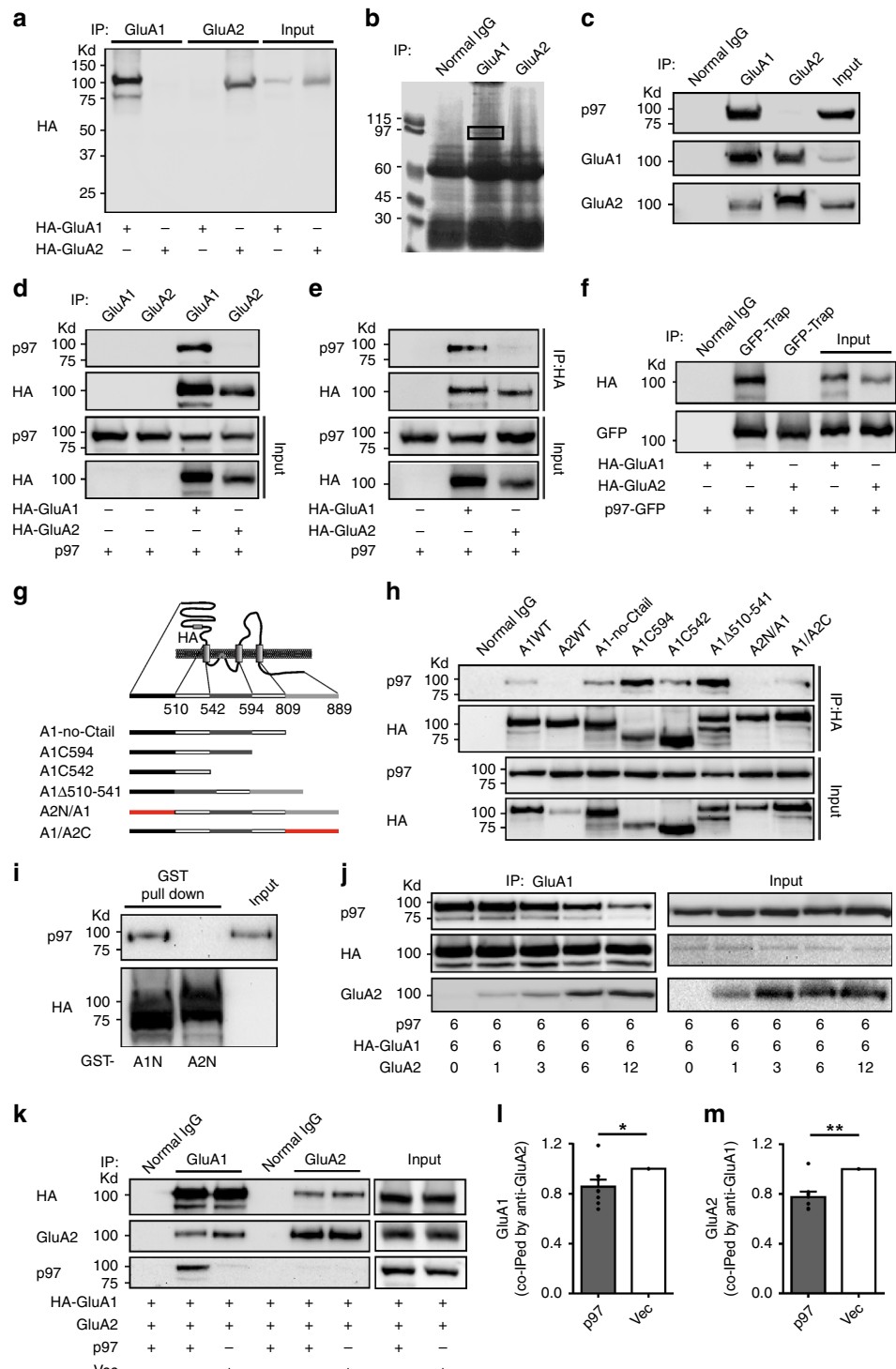

GluA2 inhibition of p97–GluA1 interaction. The decreased interaction was not an alteration in the expression levels of p97 or GluA1, because immunoblots confirmed that their expression levels were not affected by the co-expression of GluA2 (Fig. 1j). We then co-transfected equal amounts of HA-tagged GluA1 and non-tagged GluA2 plasmids with p97 or control vector. Sequential immunoblotting of anti-GluA1 and anti-GluA2 Co-IPs showed that p97 significantly decreased the ability of GluA1 and GluA2 to co-immunoprecipitate each other (Fig. 1k–m). These results strongly suggest that there is a mutually competitive inhibition between p97 and GluA2 for interacting with GluA1.

Thus, by specifically interacting with GluA1, p97 promotes the formation of GluA1-homo AMPARs at the expense of reducing the GluA1/GluA2 heteromeric AMPARs.

**p97 reduces surface expression of GluA1-homo AMPARs.** It is noteworthy that although native GluA1-homo AMPARs are thought to be expressed at least in the hippocampal pyramidal neurons[6], recent electrophysiological studies have failed to detect them functionally under basal conditions[8,9]. This may suggest that GluA1-homo AMPARs are usually intracellularly localized

**Fig. 1** p97 specifically interacts with and regulates the formation of GluA1-homo AMPARs. **a** Immunoblot reveals the subunit specificity of anti-GluA1 and anti-GluA2 in immunoprecipitation of respective subunits transiently expressed in COS7 cells. **b** Identification of p97 as a protein specifically present in GluA1, but not GluA2, complexes immunoprecipitated from the adult rat hippocampal homogenates. The gel was stained by Coomassie Blue, and the black rectangle indicates the gel area cut for mass spectrometric analysis. **c** p97 specifically complexes with the GluA1, but not GluA2, subunit in rat hippocampal tissue lysates. **d–f** p97 specifically interacts with GluA1 in recombinant expression COS7 cells. COS7 cells were transfected with p97 and HA-GluA1 or HA-GluA2 (**d**, **e**), or p97-GFP and HA-GluA1 or HA-GluA2 (**f**), and immunoprecipitated with anti-GluA1 and anti-GluA2 (**d**), anti-HA (**e**), or GFP-Trap (**f**) antibodies. Sequential immunoblots reveal that p97 can only be co-immunoprecipitated with co-expressed HA-GluA1, but not HA-GluA2. **g** Schematic diagrams illustrate the putative domain structure of GluA1 (top panel) and GluA1 mutant constructs of various domain deletions or swaps (bottom panel). **h** Immunoblotting of anti-HA co-immunoprecipitates identify the N-terminal of GluA1 as the interacting domain for p97. The interaction of p97 with GluA1 was abolished when the GluA1 N-terminal was swapped by GluA2 N-terminal. **i** In vitro recombinant protein binding assays between GST-GluA1NT or GST-GluA2NT and p97 reveal the direct interaction between p97 and the N-terminal of GluA1. **j–m** p97 promotes the formation of GluA1-homo AMPARs by inhibiting the formation of GluA1/GluA2 heteromeric AMPARs in COS7 cells. Reciprocal immunoprecipitation of the same samples and sequential immunoblotting with antibodies recognizing HA-GluA1 and GluA2 were used to determine the formation of GluA1 and GluA2 complexes. Increasing amount of GluA2 plasmids reduced the amount of p97 associated with GluA1 (**j**); and p97 reduced the ability of GluA1 or GluA2 to precipitate each other (**k–m**; $n = 8$ repeated experiments, $*p < 0.05$, $**p < 0.01$, two-tailed $t$ test). The error bars represent SEM

### Table 1 Peptide fragments of p97 in the anti-GluA1 immunoprecipitates identified by mass spectrometric analysis

| Peptide sequence | Peptide mass | m/z | Residues |
|---|---|---|---|
| LIVDEAINEDNSVVSLSQPK | 2169.12 | 2170.12 | 26–45 |
| MDELQLFR | 1050.52 | 1051.52 | 46–53 |
| GDTVLIK | 744.44 | 745.43 | 54–60 |
| NNLRVR | 770.45 | 771.43 | 90–95 |
| LGDVISIQPCPDVK | 1539.80 | 1540.81 | 96–109 |
| KGDIFLVR | 946.56 | 947.57 | 148–155 |
| GDIFLVR | 818.46 | 819.48 | 149–155 |
| EDEEESLNEVGYDDIGGCR | 2184.88 | 2185.94 | 192–210 |
| EMVELPLR | 985.53 | 986.54 | 218–225 |
| GILLYGPPGTGK | 1171.66 | 1172.66 | 240–251 |
| AVANETGAFFFLINGPEIMSK | 2255.13 | 2256.15 | 257–277 |
| LAGESESNLR | 1074.53 | 1075.54 | 278–287 |
| KAFEEAEK | 950.47 | 951.47 | 288–295 |
| NAPAIIFIDELDAIAPK | 1809.99 | 1810.98 | 296–312 |
| IVSQLLTLMDGLK | 1429.82 | 1430.83 | 324–336 |
| FGRFDR | 796.40 | 797.39 | 360–365 |
| EVDIGIPDATGR | 1241.63 | 1242.63 | 366–377 |
| LEILQIHTK | 1093.65 | 1094.65 | 378–386 |
| WALSQSNPSALR | 1328.68 | 1329.69 | 454–465 |
| ETVVEVPQVTWEDIGGLEDVKR | 2497.27 | 2498.28 | 466–487 |
| ELQELVQYPVEHPDK | 1822.91 | 1823.92 | 488–502 |
| ELQELVQYPVEHPDKFLK | 2211.16 | 2212.16 | 488–505 |
| FGMTPSK | 766.37 | 767.38 | 506–512 |
| GVLFYGPPGCGK | 1250.61 | 1251.61 | 513–524 |
| AIANECQANFISIK | 1577.79 | 1578.80 | 530–543 |
| GPELLTMWFGESEANVR | 1950.91 | 1951.92 | 544–560 |
| QAAPCVLFFDELDSIAK | 1922.94 | 1923.93 | 568–584 |
| GGNIGDGGGAADR | 1115.50 | 1116.50 | 587–599 |
| VINQILTEMDGMSTK | 1678.83 | 1679.84 | 600–614 |
| VINQILTEMDGMSTKK | 1806.92 | 1807.91 | 600–615 |
| LDQLIYIPLPDEK | 1555.85 | 1556.85 | 639–651 |
| DVDLEFLAK | 1048.54 | 1049.54 | 669–677 |
| MTNGFSGADLTEICQR | 1798.80 | 1799.80 | 678–693 |
| ESIESEIR | 961.47 | 962.47 | 701–708 |
| ESIESEIRR | 1117.57 | 1118.57 | 701–709 |
| QTNPSAMEVEEDDPVPEIR | 2154.97 | 2155.99 | 714–732 |
| RDHFEEAMR | 1189.53 | 1190.55 | 733–741 |
| RSVSDNDIR | 1060.53 | 1061.53 | 745–753 |
| SVSDNDIR | 904.42 | 905.44 | 746–753 |
| SVSDNDIRK | 1032.52 | 1033.53 | 746–754 |
| KYEMFAQTLQQSR | 1628.80 | 1629.81 | 754–766 |

*Note*: Mascot score for p97: 240; peptide sequence coverage: 54%

and therefore nonfunctional. We, therefore, next examined if p97, through its specific interaction with GluA1, also plays critical roles in regulating the trafficking and cell surface expression of GluA1-homo AMPARs. We used a combination of electrophysiological and biochemical methods in HEK293 cells transiently expressing AMPARs of various subunit combinations with or without the co-expression of p97 (Fig. 2a–f). We induced AMPAR-gated whole-cell currents with fast perfusions of 50 μM kainic acid at various holding membrane potentials and then constructed $I–V$ curves. As shown in Fig. 2a–c, we found that depending on the subunits transfected in the cells, AMPAR-gated currents exhibited differentially rectifying $I–V$ relationships. Currents recorded in cells expressing GluA1 or GluA2 alone exhibited more pronounced rectification, with GluA1 being more inwardly and GluA2 more outwardly rectified. In contrast, currents from cells co-transfected with GluA1 and GluA2 along with control vector had a more linear $I–V$ curve. These results are greatly in-line with the known critical role of the GluA2 subunit in determining AMPAR current rectification[3,5]. Interestingly, co-transfection of GluA1 and GluA2 along with p97 resulted in a much more outwardly rectifying $I–V$ curve, with an apparent $I–V$ curve being shifted toward that of GluA2 alone (Fig. 2b, c). More pronouncedly, addition of p97 also drastically reduced the amplitude of the whole-cell currents in comparison with currents from cells only expressing GluA1 and GluA2 (Fig. 2a, d).

Considering the potential role of p97 in promoting the formation of GluA1-homo AMPARs at the expense of reducing the formation of the GluA1/GluA2 heteromeric (Fig. 1k–m), and that GluA1-homo AMPARs have a more inward rectification than the GluA2-containing AMPARs[3,5], we reasoned that the outward shift of $I–V$ curve in the presence of p97 is likely a result of the reduced cell surface expression of GluA1-homo AMPARs. To test this prediction, we analyzed the surface expression of the GluA1 and GluA2 subunits using surface biotinylation in HEK293 cells co-transfected with p97 or control vector as well as GluA1 and GluA2. As shown in Fig. 2e, f, co-expression of p97 significantly reduced the level of GluA1, but not GluA2, on the cell surface without altering the levels of total GluA1 or GluA2.

To assess the role of p97 in modulating cell surface expression of native AMPARs, we overexpressed p97 in cultured hippocampal neurons, followed by immunostaining surface GluA1 (Fig. 2g) or GluA2 (Fig. 2h) and vesicular glutamate transporter 1 (vGluT1), an excitatory presynaptic marker. Consistent with the reduction in cell surface expression of GluA1 in HEK293 cells (Fig. 2e, f), overexpression of p97 reduced the intensity of postsynaptic surface clusters of GluA1, but not GluA2 (Fig. 2g, h).

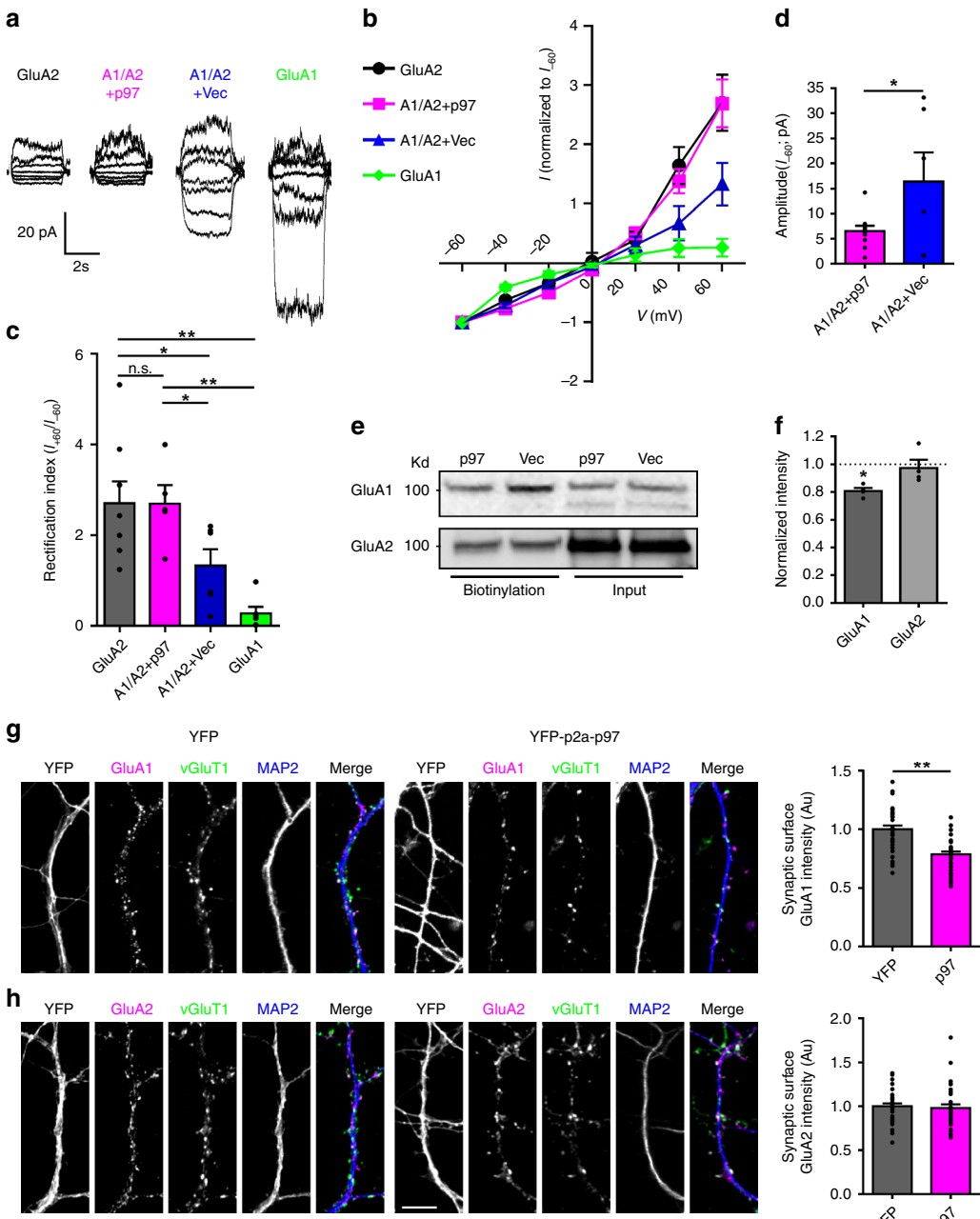

**Fig. 2** p97 regulates subcellular localization of GluA1-homo AMPARs. **a–d** HEK293 cells were transfected with GluA1, GluA2, GluA1/GluA2 plus p97 (A1/A2 + p97), or GluA1/GluA2 plus control vector (A1/A2 + Vec). AMPAR-mediated whole-cell currents were induced by fast application of kainite (50 μM; to reduce receptor desensitization) while holding membrane potentials from −60 to +60 mV with a step interval of 20 mV. Representative traces and I–V curves are shown in (**a**) and (**b**), respectively, and rectification indexes ($I_{+60/−60~mV}$) in (**c**) are quantified from recordings in (**b**) (GluA2: $n = 8$ cells; A1/A2 + p97: $n = 5$ cells; A1/A2 + Vec: $n = 6$ cells; GluA1: $n = 6$ cells; $F = 8.843$, $p = 0.0006$, one-way ANOVA; *$p < 0.05$, **$p < 0.01$, post hoc test). **d** Bar graphs of the peak current amplitude recorded at the holding potential of −60 mV show that overexpression of p97 reduces whole-cell currents (A1/A2 + p97: $n = 10$ cells; A1/A2 + Vec: $n = 6$ cells; *$p < 0.05$, two-tailed $t$ test). **e, f** Immunoblots sequentially probed for GluA1 and GluA2 following surface biotinylation in HEK293 cells reveal that co-expression of p97 with GluA1 and GluA2 subunits specifically reduces the surface expression of GluA1, but not GluA2, subunits ($n = 4$ repeated experiments; *$p < 0.05$, two-tailed $t$ test, compared with Vec). **g, h** Cultured hippocampal neurons were transfected with YFP or YFP-p2a-p97 (p97) at 0 DIV. Neurons were immunostained live using anti-GluA1 (**g**) or GluA2 (**h**) antibody under nonpermeable conditions at 14 DIV, followed by fixation and immunostaining for VGluT1 and the dendritic marker MAP2 under permeable conditions. Overexpression of p97 significantly reduced synaptic surface GluA1 intensity without changing GluA2 intensity (scale bar represents 10 μm; $n = 33$ cells from two independent experiments, **$p < 0.01$, two-tailed $t$ test). The error bars represent SEM

The results that p97 does not increase, but actually reduces the GluA1 subunit expression on the cell surface raise the question of whether p97 not only affects the formation of GluA1-homo AMPARs, but also modulates their trafficking. To investigate this further, we co-transfected HA-GluA1 with p97 or control vector into COS7 cells, and directly analyzed GluA1-homo AMPARs expression on cell surface and intracellular compartments using both biotinylation and immunostaining. As expected,

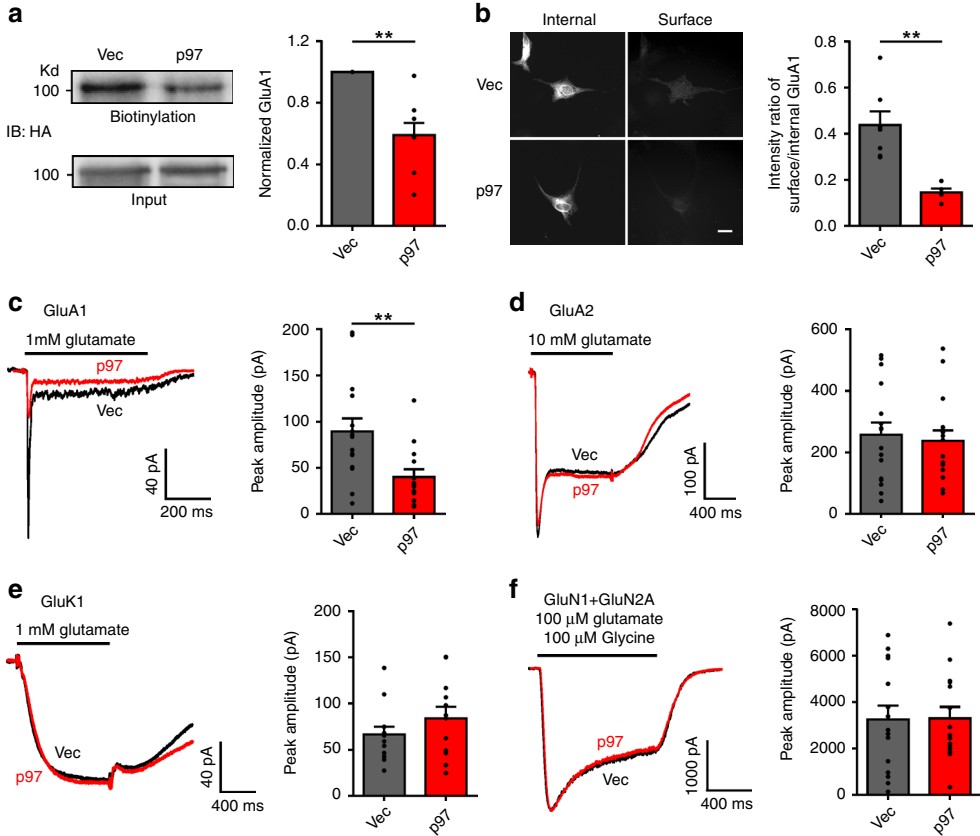

**Fig. 3** p97 decreases cell surface recombinant GluA1-homo AMPARs. **a**, **b** p97 reduces surface expression of GluA1-homo AMPARs in COS7 cells. HA-GluA1 was transfected with vector (Vec) or p97. Surface expression of GluA1-homo AMPARs were assayed with surface biotinylation (**a**; $n = 6$ repeated experiments, **$p < 0.01$, two-tailed $t$ test) or immunofluorescent microscopic examination following sequential nonpermeabilized (Surface) and permeabilized (Internal) staining (**b**; Vec: $n = 8$ coverslips; p97: $n = 6$ coverslips; **$p < 0.01$, two-tailed $t$ test; Scale bar represents 20 μm). **c–f** Whole-cell currents were induced with fast application of saturating concentration of agonizts on HEK293 cells transfected with HA-GluA1 (**c**), GluA2 (**d**), GluK1 (**e**), or GluN1 + GluN2A (**f**) and control vector (Vec) or p97, along with GFP to detect the transfected cells for recording. p97 decreased the peak amplitude of GluA1-homo receptor-mediated currents (**c**; Vec: $n = 15$ cells; p97: $n = 14$ cells; **$p < 0.01$, two-tailed $t$ test), without changing that of GluA2 homomeric receptor (**d**; Vec: $n = 16$ cells; p97: $n = 16$ cells), GluK1 homomeric receptor (**e**; Vec: $n = 13$ cells; p97: $n = 12$ cells), or GluN1 + GluN2A receptor (**f**; Vec: $n = 15$ cells; p97: $n = 15$ cells) mediated currents. The error bars represent SEM

overexpression of p97 significantly reduced the surface level of GluA1 without altering the total level as demonstrated by biotinylation (Fig. 3a); and immunofluorescent imaging (Fig. 3b) confirmed that p97, while promoting their formation, retained GluA1-homo AMPARs intracellularly.

We next studied whether p97 affects GluA1-homo AMPARs-mediated currents by fast perfusion of glutamate (1 mM) in HEK293 cells transiently co-expressing GluA1 and p97 or the control vector. Consistent with the reduced surface expression level of GluA1, co-transfection of p97 significantly reduced the amplitude of peak currents (Fig. 3c). We also tested the functional specificity of p97 modulation of GluA1-homo AMPARs by comparing its effect on the GluA2 homomeric receptors (Fig. 3d), Kainate receptors (GluK1 homomeric; Fig. 3e), and NMDA receptors (GluN1 + GluN2A; Fig. 3f). In contrast with the reduction of GluA1-homo receptor currents, p97 did not change the currents through the GluA2 homomeric receptor, GluK1 homomeric receptor, or GluN1/GluN2A NMDA receptors (Fig. 3d–f). Taken together, the results demonstrate that p97 promotes the formation of GluA1-homo AMPARs and retains a large portion of them intracellularly.

**p97 regulates GluA1-homo AMPARs in cultured neurons**. As discussed above, previous studies have provided some evidence

that a small quantity of GluA1-homo AMPARs are expressed in the hippocampal neurons and can be rapidly translocated into the postsynaptic plasma membrane during the expression of hippocampal CA1 LTP[4,8] (but also see ref. [9]). Since p97 interacts with and retains the GluA1 subunits intracellularly, we next investigated if p97 plays a critical role in regulating the intracellular retention of GluA1-homo AMPARs under basal conditions and their rapid release during LTP production in hippocampal neuron cultures. Since a previous study has reported that transient transfection of GluA1 in cultured hippocampal neurons may predominantly form homomeric AMPARs through their preferred self-dimerization[14], we first investigated if p97 can retain these GluA1-homo AMPARs under basal conditions using neurons that were transfected with GluA1 and p97 or the control vector. Miniature excitatory postsynaptic currents (mEPSCs) were recorded, and their component mediated by GluA1-homo AMPARs was inferred by the sensitivity of mEPSCs to the GluA2-lacking AMPAR antagonist philanthotoxin-433 (PhTx, 10 μM)[8,14]. We found that in neurons transfected with GluA1 and the control vector, PhTx significantly decreased the frequency and amplitude of mEPSCs (Fig. 4a–e), suggesting the presence of a substantial amount of GluA1-homo AMPARs. However, in neurons co-transfected with GluA1 and p97, the ability of PhTx to decrease both the frequency and amplitude of mEPSCs was

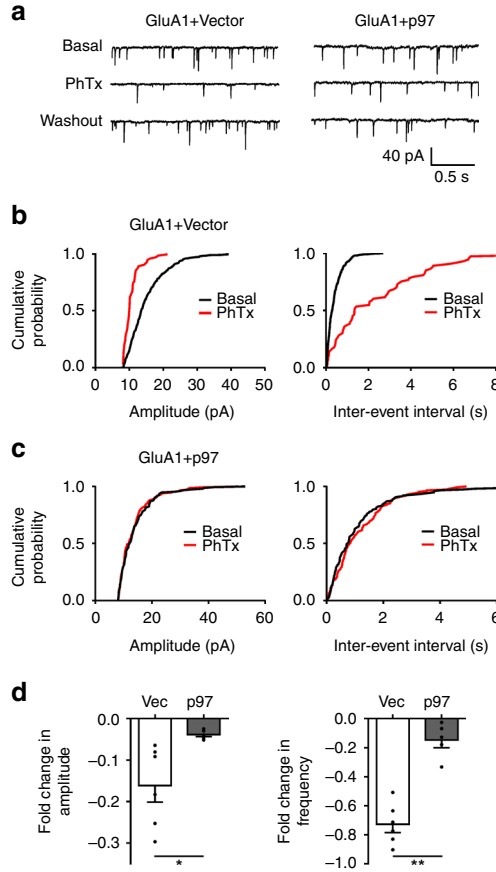

**Fig. 4** p97 decreases recombinant GluA1-homo AMPAR-mediated currents in cultured hippocampal neurons. Hippocampal neurons in primary cultures were transfected with HA-GluA1 and control vector (GluA1 + Vec) or p97 (GluA1 + p97), along with GFP to detect the transfected cells for recording. mEPSCs were recorded under whole-cell voltage-clamp at a holding membrane potential of −60 mV 48 h after transfection. Surface expression of recombinant of GluA1-homo AMPARs was inferred with the presence of the sensitive component of mEPSCs to bath application of PhTx (10 μM). From **a–e** are represented traces (**a**), cumulative probability plots of amplitudes and inter-event intervals (**b, c**), and bar graphs of averaged fold changes in amplitude (**d**) and frequency (**e**) of mEPSCs recorded 5 min before (Basal) and 5 min after (PhTx) PhTx application (GluA1 + Vec: $n = 6$ cells; GluA1 + p97: $n = 5$ cells; $*p < 0.05$, $**p < 0.01$, two-tailed $t$ test). The error bars represent SEM

abolished (Fig. 4a–e), suggesting a lack of detectable level of GluA1-homo AMPARs on the plasma membrane due to the intracellular retention by p97 overexpression. These results are consistence with the results we observed in the recombinant expression cells, and further support that p97 plays an important role in retaining the newly formed GluA1-homo AMPARs in the intracellular reserve pool.

To test whether this p97-dependent reserve pool of GluA1-homo AMPARs has a critical role in mediating the rapid insertion of the native GluA1-homo AMPARs following LTP induction observed in previous studies[8,14], we compared effects of the PhTx-sensitive component of mEPSCs before and 5 min after the induction of LTP using a well-established glycine-induced chemical LTP model in cultured hippocampal neurons[15]. As shown in Fig. 5a–e, application of PhTx produced little effect on the basal mEPSCs, providing further evidence for the absence of surface GluA1-homo AMPARs under basal conditions[8,16]. A brief application of glycine (200 μM; 3 min) reliably induced LTP as

previously reported[15] (Fig. 5a–e). Importantly, this glycine induced increase of mEPSCs was significantly decreased by PhTx (Fig. 5a–e). This PhTx sensitive component of mEPSCs emerges shortly after LTP induction, which strongly suggests that, similar to that observed in acute hippocampal slices[8] (but also see ref.[9]), there is a rapid plasma membrane insertion of GluA1-homo AMPARs at the potentiated synapses shortly after LTP induction in cultured hippocampal neurons under our experimental conditions.

Since p97 plays a critical role in maintaining an intracellular reserve pool of GluA1-homo AMPARs, we reasoned that there must be a rapid dissociation of p97–GluA1 upon stimulation with LTP production protocol. The dissociation would then facilitate the release and delivery of these GluA1-homo AMPARs from the intracellular reserve pool to the postsynaptic membrane, and hence enabling the expression of LTP. We tested the association between p97 and GluA1 using Co-IP prior to (time 0) and at various time points after the induction of LTP. Sequential immunoblotting of anti-GluA1 immunoprecipitates revealed that there was a time-dependent and transient decrease in p97–GluA1 association within 30 min of the LTP induction (Fig. 5f, g). The dissociation was specific to LTP-producing stimulation protocol as it was not detected following a stimulation with an established chemical LTD-inducing protocol[15,17] (NMDA 10 μM; 3 min; Fig. 5h, i). We also tested whether increasing p97 expression levels affects glycine induced LTP. In cultured hippocampal neurons transfected with p97, glycine (200 μM; 3 min) application failed to induce LTP; whereas glycine induced long-lasting potentiation of mEPSC in the control neurons (Supplementary Fig. 1a-e). Our results suggest that through the association with GluA1, p97 plays a critical role in regulating the formation and trafficking of GluA1-homo AMPARs, and hence LTP expression.

**p97 modulates GluA1-homo AMPARs and LTP in brain slices.** To further validate the critical role of p97 in regulating the trafficking and expression of GluA1-homo AMPARs and LTP expression in a more physiological context, we examined the effects of p97 on mEPSCs and electrical stimulation-induced LTP at the CA1 synapses in hippocampal slices. We either increased p97 activity through AAV-delivered overexpression of p97 or decreased its activity with DBeQ, a specific p97 inhibitor[18]. We did not use viral shRNA knockdown or activity-dead dominant p97 because these manipulations invariably caused neuronal death in our preliminary results. Prior to using DBeQ in electrophysiological experiments, we first determined if inhibiting p97 activity with DBeQ can disrupt the p97–GluA1 interaction, and thereby affect cell surface expression of GluA1-homo AMPARs. As expected, DBeQ treatment (11 μM) not only significantly reduced the p97–GluA1 interaction (Fig. 6a, b), but also significantly increased the cell surface expression level of GluA1 in cultured hippocampal neurons (Fig. 6c, d). Interestingly, the dissociation of p97–GluA1 by DBeQ treatment is dose-dependent, with an $IC_{50}$ of 1.09 μM, and the dose (11 μM) we used in the following electrophysiological experiments caused a maximal inhibition of p97–GluA1 interaction (Fig. 6e, f). These results validated DBeQ as a useful tool in disrupting p97–GluA1 interaction, and hence releasing GluA1-homo AMPARs from the intracellular reserve pool to the plasma membrane.

Consistent with the results observed in neuronal cultures, we found that intracellular application of DBeQ (11 μM) through the recording pipette into CA1 neurons resulted in a gradual increase in the mEPSC frequency, with the number of events recorded in the fourth 5 min being significantly larger than that of the first 5 min recording (Fig. 6g–k). The DBeQ-induced increase in mEPSC frequency was a result of increased postsynaptic

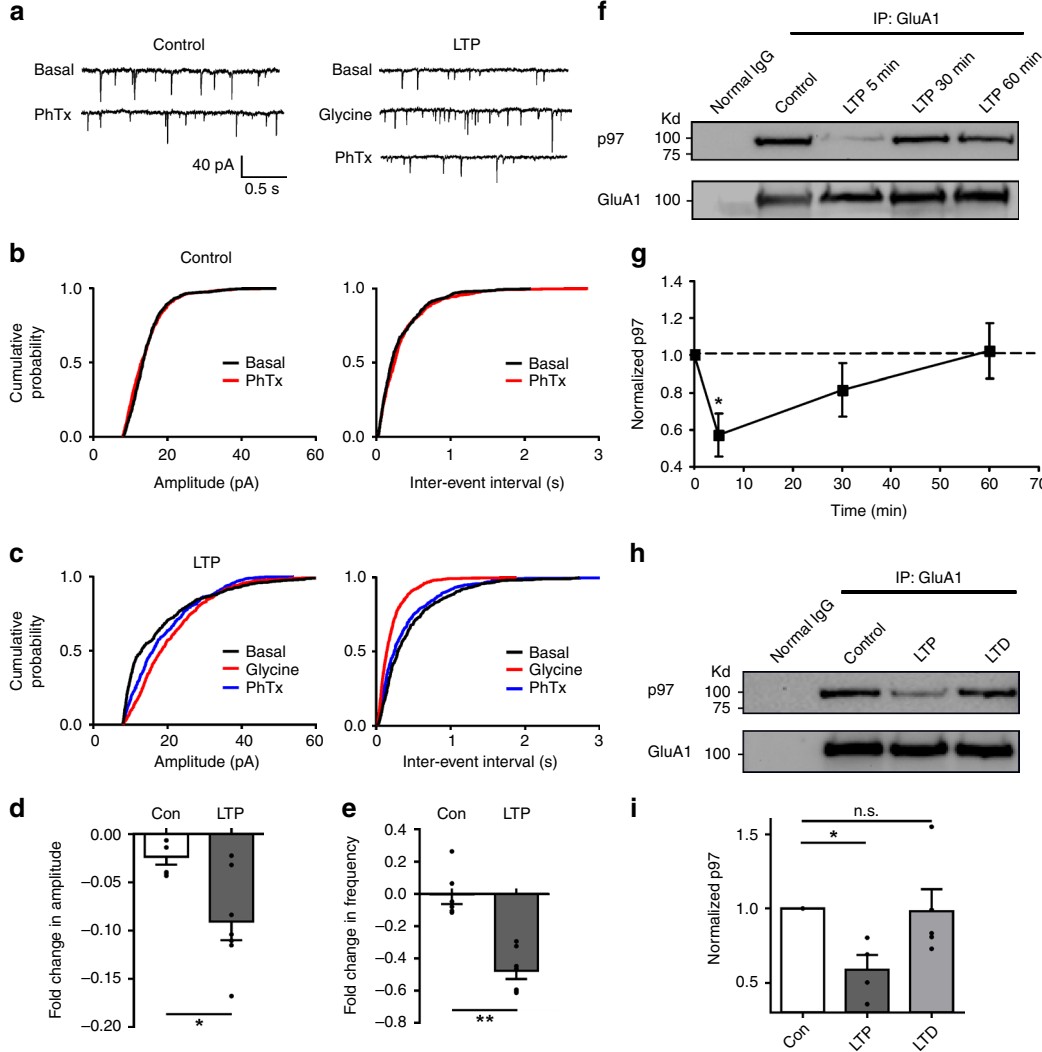

**Fig. 5** Synaptic insertion of native GluA1-homo AMPARs is associated with a rapid dissociation of p97 during LTP in cultured hippocampal neurons. **a–c** Representative traces (**a**) and cumulative probability plots of the amplitude and inter-event intervals of mEPSCs (**b**, **c**) from individual neurons in the absence and presence of PhTx show that bath application of glycine (200 μM, 3 min) produced chemical LTP as evident by the increased amplitude and frequency of mEPSCs (LTP) in comparison with that of control (Control). Bath application of PhTx (10 μM; 5 min after the formation of whole-cell configuration (basal) or glycine application (LTP)), while having no observable effect on basal mEPSCs (**b**; Control; PhTx), reduced the LTP-induced increase in both amplitude and frequency of mEPSCs (**c**; LTP; PhTx), suggesting that there was a rapid insertion of the native GluA1-homo AMPARs following the LTP induction. **d**, **e**. Bar Graphs of grouping data from six to seven neurons demonstrate that PhTx only reduced both amplitude (**d**) and frequency (**e**) of mEPSCs after LTP, but not at the control (Con) conditions (Control: $n = 6$ cells; LTP: $n = 7$ cells; *$p < 0.05$, **$p < 0.01$, two-tailed $t$ test). **f–i** Sequential immunoblotting of p97 and GluA1 following co-IP with anti-GluA1 antibodies show that there was a transient and time-dependent dissociation of p97 from GluA1 immediately following the induction of LTP (**f**, **g**, $n = 6$ repeated experiments, $F = 4.256$, $p = 0.0147$, one-way ANOVA, *$p < 0.05$, post hoc test), but not after induction of LTD (**h**, **i**, 10 min after induction of LTP or LTD, $n = 5$ repeated experiments; $F = 4.582$, $p = 0.0357$, one-way ANOVA, *$p < 0.05$, n.s. not significant, post hoc test). The error bars represent SEM

incorporation of GluA1-homo AMPARs as it was prevented by the addition of PhTx in the bath solution (10 μM; Fig. 6g–k). To provide further support for a postsynaptic locus for the increased mEPSC frequency, we analyzed the ratios of the AMPA/NMDA components of the evoked EPSCs in the presence or absence of DBeQ. As shown in Fig. 7a, DBeQ significantly increased the AMPA/NMDA ratio; and this increase again was reversed by the addition of PhTx. In addition, application of DBeQ changed postsynaptic AMPAR rectification as shown by an inward shift of I–V relation curve of evoked AMPAR mediated EPSC (Fig. 7c–e). Thus, similar to the results observed in neuronal cultures, p97 appears to play a critical role in maintaining an intracellular reserve pool of the native GluA1-homo AMPARs. Disrupting the

p97–GluA1 interaction with DBeQ is sufficient to release these receptors to the plasma membrane.

To directly test if p97 is involved in mediating the previously reported plasma membrane insertion of GluA1-homo AMPARs during the initial phase of LTP expression[4,8], we next induced LTP at the CA1 synapse using standard paired stimulation protocol under whole-cell recording with or without DBeQ (11 μM) in the recording pipette[19]. In cells recorded without DBeQ, the amplitudes of evoked EPSCs prior to the LTP induction were constant. Application of the paired stimulation reliably induced LTP that lasted throughout the recording period of more than 40 min (Fig. 7b). In contrast, in cells recorded with pipettes containing DBeQ, there was a gradual increase in the amplitude

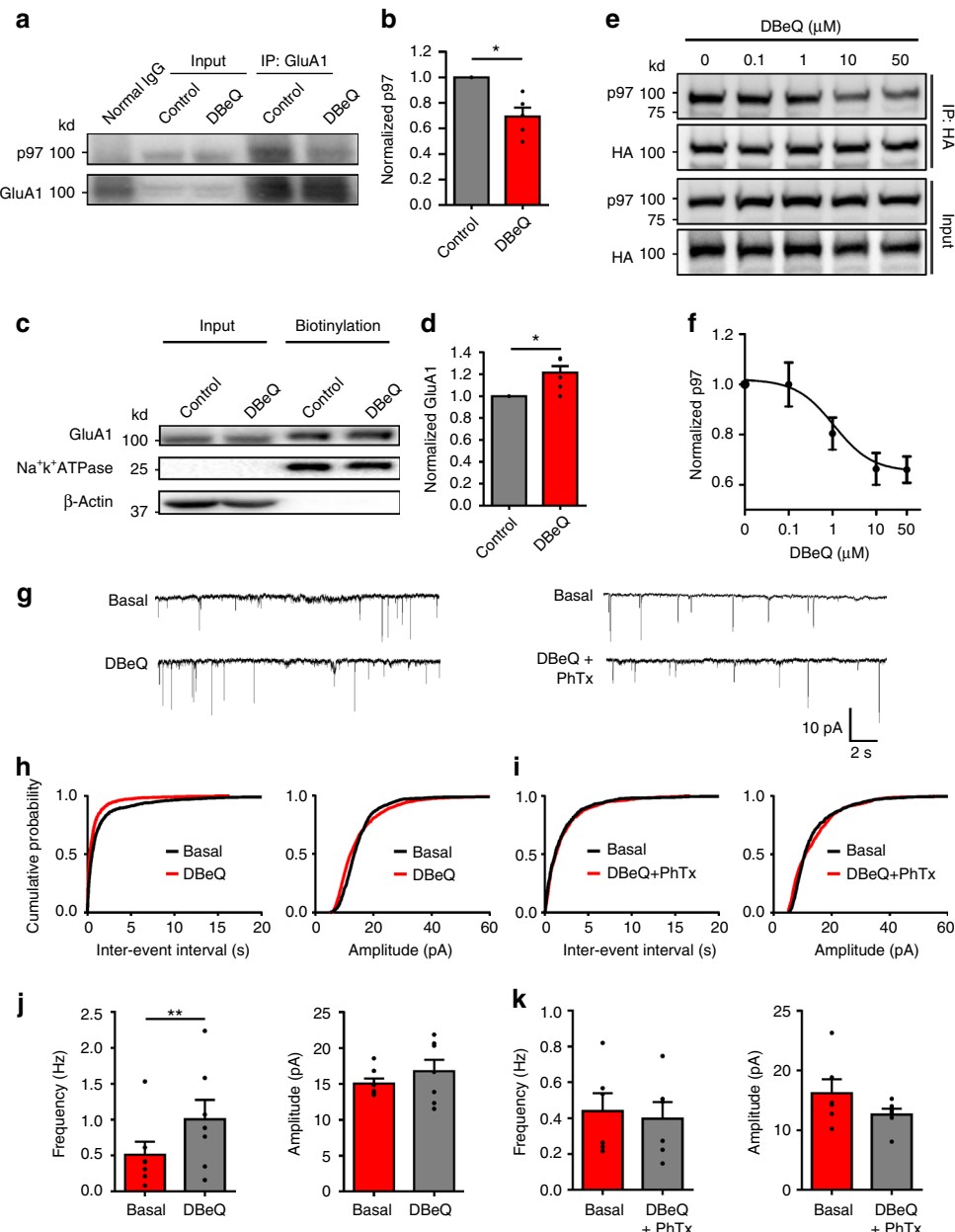

**Fig. 6** p97 modulates native GluA1-homo AMPAR trafficking in hippocampal slices. **a–d** Neuronal cultures were treated with DBeQ in the bath (11 μM; 30 min). Sequential immunoblotting for p97 and GluA1 in the anti-GluA1 co-immunoprecipitates (**a**) and bar graph of grouping data (**b**; $n = 5$ repeated experiments) demonstrate that inhibition of p97 with DBeQ significantly decreased the p97-GluA1 interaction; sequential immunoblots following biotinylation for surface GluA1 and Na+K+-ATPase, with β-actin as an intracellular protein control (**c**) and bar graph of grouping data (**d**; $n = 6$ repeated experiments) reveal that disrupting the p97–GluA1 interaction by DBeQ released GluA1-homo AMPARs from intracellular reserve pool into the plasma membrane of neurons (*$p < 0.05$, two-tailed $t$ test). **e**, **f** DBeQ disrupts p97-GluA1 interaction. COS7 cells were transfected with p97 and HA-GluA1, treated with different doses of DBeQ (in DMEM; 0, 0.1, 1, 10, 50 μM; 30 min), and co-immunoprecipitated by anti-HA antibody. Sequential immunoblotting for p97 and GluA1 (**e**) and dose-response curve (**f**) of grouping data show that DBeQ dissociates p97–GluA1 interaction in a dose-dependent manner, with an IC50 of 1.09 μM ($n = 5$ repeated experiments). **g–k** mEPSC recordings reveal that intracellular application of p97 inhibitor DBeQ (11 μM) through the recording pipette significantly increased mEPSC frequency, and the effect of DBeQ was reversed by bath application of GluA1-homo AMPAR inhibitor PhTx (10 μM) (DBeQ: $n = 7$ cells; DBeQ + PhTx: $n = 6$ cells; **$p < 0.01$, two-tailed paired $t$ test). From **g** to **k** are represented traces (**g**), cumulative probability plots of inter-event intervals and amplitude (**h**, **i**), and bar graphs of averaged frequencies and amplitudes of mEPSCs recorded 0–5 min (Basal) and 15–20 min (DBeQ) (**j**) or (DBeQ + PhTx) (**k**) after the formation of the whole-cell configuration. The error bars represent SEM

of evoked EPSCs that usually plateaued at a stable level within 15 min after the formation of the whole-cell configuration. Moreover, delivering paired stimulation after the potentiation plateaued failed to induce LTP (Fig. 7b). DBeQ induced run-up of evoked EPSCs is specific to AMPAR mediated eEPSCs, as we did not observe any change in NMDAR-mediated eEPSCs under

the same condition (Supplementary Fig. 2). To test whether the run-up of evoked EPSCs is due to the synaptic insertion of GluA1-homo AMPARs, we bath applied PhTx (10 μM) immediately or 20 min after the formation of whole-cell configuration. PhTx completely blocked the run-up of evoked EPSCs when applied immediately (Fig. 7f); and it reversed the potentiated

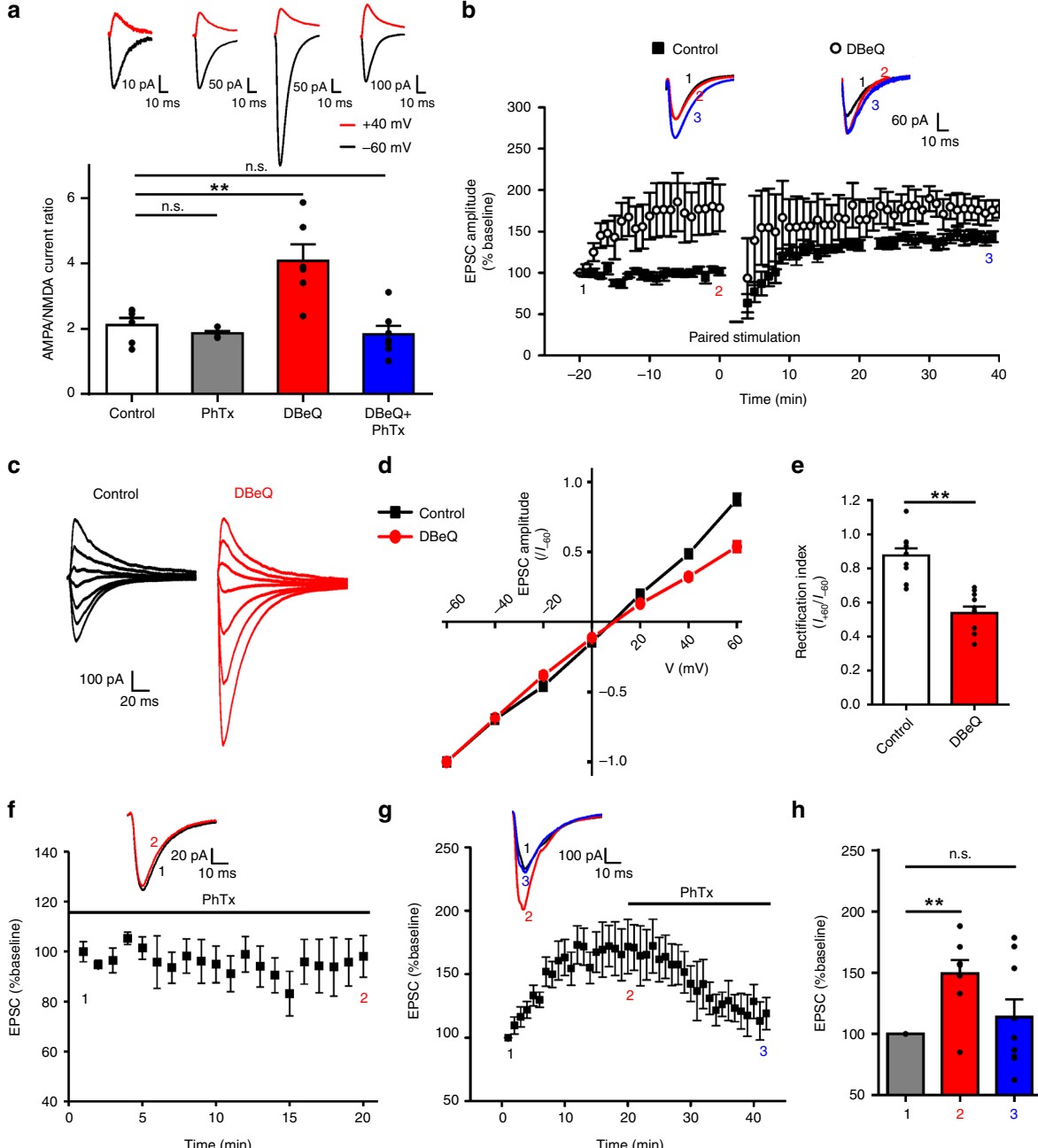

**Fig. 7** Inhibition of p97 promotes native GluA1-homo AMPAR postsynaptic insertion in hippocampal slices. **a** Inhibition of p97 increases AMPA/NMDA ratio. Twenty minute after the formation of whole-cell configuration, AMPAR- and NMDAR-mediated EPSCs were evoked by electrical stimulation of schaffer collateral inputs at holding potentials of −60 and +40 mV, respectively. The AMPA/NMDA ratio was significantly increased by DBeQ (11 μM; through the recording pipette) in the absence of PhTx, but not in the presence of PhTx (10 μM; bath application throughout the recording) (Control: $n = 6$ cells; PhTx: $n = 4$ cells; DBeQ: $n = 6$ cells; DBeQ + PhTx: $n = 7$ cells; $F = 11.01$, $p = 0.0002$, one way ANOVA; **$p < 0.01$, n.s. not significant, post hoc test). **b** Inhibition of p97 increases basal synaptic transmission, and occludes the induction of LTP. Paired stimulation protocol reliably induced LTP in neurons recorded with pipettes without DBeQ (Line with black squares; $n = 6$ cells). DBeQ (11 μM) contained in recording pipettes increased basal EPSCs that prevented paired stimulation to produce LTP (Line with empty circles; $n = 4$ cells). **c–e** Inhibition of p97 changes the rectification of AMPAR EPSCs. Twenty minute after the formation of whole-cell configuration, AMPAR mediated EPSCs were evoked by electrical stimulation of schaffer collateral inputs at holding potentials from −60 to +60 mV with a step interval of 20 mV. DBeQ (11 μM) contained in recording pipettes shifted I-V curve to a more inward rectification, as shown in representative traces (**c**), I–V curve (**d**), and rectification index ($I_{+60/-60 \text{ mV}}$; **e**) (Control: $n = 10$ cells; DBeQ: $n = 10$ cells; **$p < 0.01$, two-tailed t test). **f** Bath application of PhTx (10 μM) immediately after the formation of whole-cell configuration blocked the run-up of AMPAR mediated EPSCs ($n = 8$ cells). **g**, **h** PhTx (10 μM), bath applied 20 min after the formation of whole-cell configuration when AMPAR mediated EPSCs reached the plateau, reversed the EPSC run-up, decreasing the EPSC amplitude back to the baseline level within 20 min of PhTx application ($n = 9$ cells; $F = 5.842$, $p = 0.0089$, one-way ANOVA; **$p < 0.01$, n.s. not significant, post hoc test). The error bars represent SEM

eEPSCs back to baseline level if applied after the plateau (Fig. 7g, h). These results suggest that DBeQ causes a rapid increase in surface expression of GluA1-homo AMPARs by inhibiting p97, which occludes the subsequent induction of LTP.

If our reasoning is correct, one can expect that overexpression of p97 would increase its ability to retain native GluA1-homo AMPARs in the intracellular compartment, thereby reducing cell surface expression of the receptor under basal conditions or following LTP induction. We tested this prediction by over-expressing p97 in hippocampal CA1 neurons by injecting AAV9-YFP-p2a-p97 or AAV9-GFP as control into the mouse ventricles at P0. As predicted, overexpression of p97, but not GFP, caused a significant decrease in the mEPSC frequency and amplitude (Fig. 8a–c). Importantly, the p97-reduced mEPSCs were fully reversed by the addition of DBeQ in the recording pipette (Fig. 8d–f). These results are likely due to the reduced postsynaptic AMPARs as a result of p97-promoted formation of intracellular GluA1-homo AMPARs. Consistent with this conjecture, intracellular application of DBeQ produced a much more pronounced gradual increase in the amplitude of evoked EPSCs in p97 overexpressed neurons (Fig. 8g). Interestingly, while paired stimulation protocol reliably induced LTP in cells overexpressing GFP, it failed to produce LTP in cells over-expressing p97 (Fig. 8g), which is consistent with results from cultured hippocampal neurons (Supplementary Fig. 1a–e). The failure to produce LTP following p97 overexpression may imply that overexpression of p97, while promoting the formation of intracellular GluA1-homo AMPARs, increased its interaction with GluA1 to a level that was not sufficiently disrupted by a LTP producing protocol. To further confirm that GluA1-homo AMPAR synaptic insertion plays an important role in LTP under our experimental conditions, we bath applied PhTx (10 μM) and recorded LTP in the control cells expressing GFP. As shown in Fig. 8i, j, LTP was blocked by PhTx. In addition, NMDAR mediated eEPSCs were not affected by p97 overexpression (Supplementary Fig. 3).

## Discussion

The subunit composition of AMPARs determines the function, subcellular localization and trafficking of the receptors, thereby being important for mediating the expression of various forms of synaptic plasticity[1,3–5]. Although the majority of AMPARs in the hippocampus are the GluA1/GluA2 or GluA2/GluA3 heteromeric receptors, a small population of the AMPARs contains only the GluA1 subunits[3,5]. These GluA1-homo AMPARs are functionally significant as they are predominantly localized intracellularly under basal conditions, but can be rapidly delivered into the plasma membrane to enable various forms of synaptic plasticity[1,3–5]. However, mechanisms underlying their formation, intracellular retention and activity-dependent translocation into the plasma membrane remain largely unknown. In the present study, we revealed a previously unappreciated mechanism that plays a critical role in these processes. We found that p97 specifically interacts with the GluA1 subunit of homomeric AMPARs, but not the GluA1 subunit that is heteromerized with the GluA2 subunit. Moreover, we showed that through this interaction, p97 promotes the formation of GluA1-homo AMPARs at the expense of reduced GluA1/GluA2 heteromeric AMPARs. p97 retains the homomeric receptors intracellularly under basal conditions and rapidly releases them into the post-synaptic membrane during LTP production. Thus, our study identifies p97 as a GluA1-specific regulator that is critically important for GluA1-homo AMPARs specific synaptic plasticity.

Evidence accumulated in recent years strongly suggests a subunit-specific role of AMPARs in determining the direction of

NMDAR-dependent synaptic plasticity at least at the hippo-campal CA1 synapse. Specifically, the presence of the GluA1 and GluA2 subunits appears to be important for the induction of LTP and LTD, respectively[1,20–25]. However, the mechanisms under-lying the subunit differential roles in LTP and LTD remain not fully understood and hotly debated[1,20–26]. p97 shares a high sequence similarity with N-ethylmaleimide sensitive factor (NSF), both of which belong to the type II AAA ATPase family and are functionally involved in protein trafficking related membrane fusion events[27–29]. NSF interacts with GluA2 and plays an important role in modulating the GluA2-containing AMPAR trafficking, plasma membrane insertion and/or stabilization, thereby mediating various forms of synaptic plasticity and behaviors in a GluA2-subunit specific manner[30–35]. In the present study, we demonstrated that unlike NSF, p97 specifically interacts with GluA1 and plays a critical role in the formation and traf-ficking of GluA1-homo AMPARs, and hence in mediating LTP in a GluA1 subunit-specific manner. Thus, the present work, along with previous studies on the role of NSF in mediating GluA2 trafficking and synaptic plasticity, strongly suggests that the type II AAA ATPases may represent a group of important regulators for controlling subunit-specific AMPARs trafficking and synaptic plasticity.

p97 appears to regulate GluA1-homo AMPARs and hence the receptor dependent synaptic plasticity through several ways. First, p97 regulates the formation of GluA1-homo AMPARs through its specific interaction with the GluA1 subunit. Since a vast majority of the native AMPARs in the mammalian brain are the hetero-meric GluA1/GluA2 AMPARs, our results that anti-p97 could only co-immunoprecipitate GluA1, but not GluA2, subunits of native AMPARs in neurons (Fig. 1b, c) indicate that p97 may only interact with the GluA1 subunits that have not been het-eromerized with GluA2. AMPAR subunits are synthesized and assembled as dimers of dimers in the rough endoplasmic reti-culum (ER)[36–38]. Dimerization of AMPAR subunits have been shown to prefer to hetero-dimerize from GluA1 dimer and GluA2 dimer over simply homo-dimerize themselves from two dimers[39]. The underlying mechanisms that regulate the formation of these hetero- or homo-tetramers remain poorly understood. Given that p97 is an ER residential protein, our results allow us to tentatively propose that p97 may interact with and induce conformational changes in the GluA1 monomer and/or dimers that favor their selves-dimerization over heteromerization with a GluA2 mono-mer and/or dimer. Indeed, in the present study, we were able to provide strong evidence to support that through its interaction with GluA1, p97 promotes the formation of GluA1-homo AMPARs by reducing the formation of the GluA1/GluA2 het-eromeric AMPARs (Fig. 1j–m).

Second, as an ER residential protein, p97 might retain GluA1-homo AMPARs as an intracellular receptor reserve pool. Although native GluA1-homo AMPARs are thought to be present in the mammalian brain, previous electrophysiological studies have not been able to detect these receptors functionally in the hippocampal pyramidal neurons under basal conditions[8,9]. However, mechanisms underlying the retention of these GluA1-homo receptors under basal conditions remain poorly under-stood. In the present study, we showed that overexpression of p97 not only significantly reduced the cell surface GluA1-containing AMPARs, but also the AMPAR-mediated currents in both the recombinant expression systems (Fig. 2a–f and Fig. 3a–c) and hippocampal neurons in cultures (Fig. 2g, h, Fig. 4), and in slices (Fig. 8). Importantly, this reduction of functional GluA1-homo AMPARs could be fully reversed by disrupting the p97–GluA1 interaction with the p97 inhibitor DBeQ (Fig. 6 and Fig. 7). Thus, our results indicate that through its interaction with the GluA1 subunit, p97 retains GluA1-homo AMPARs in the

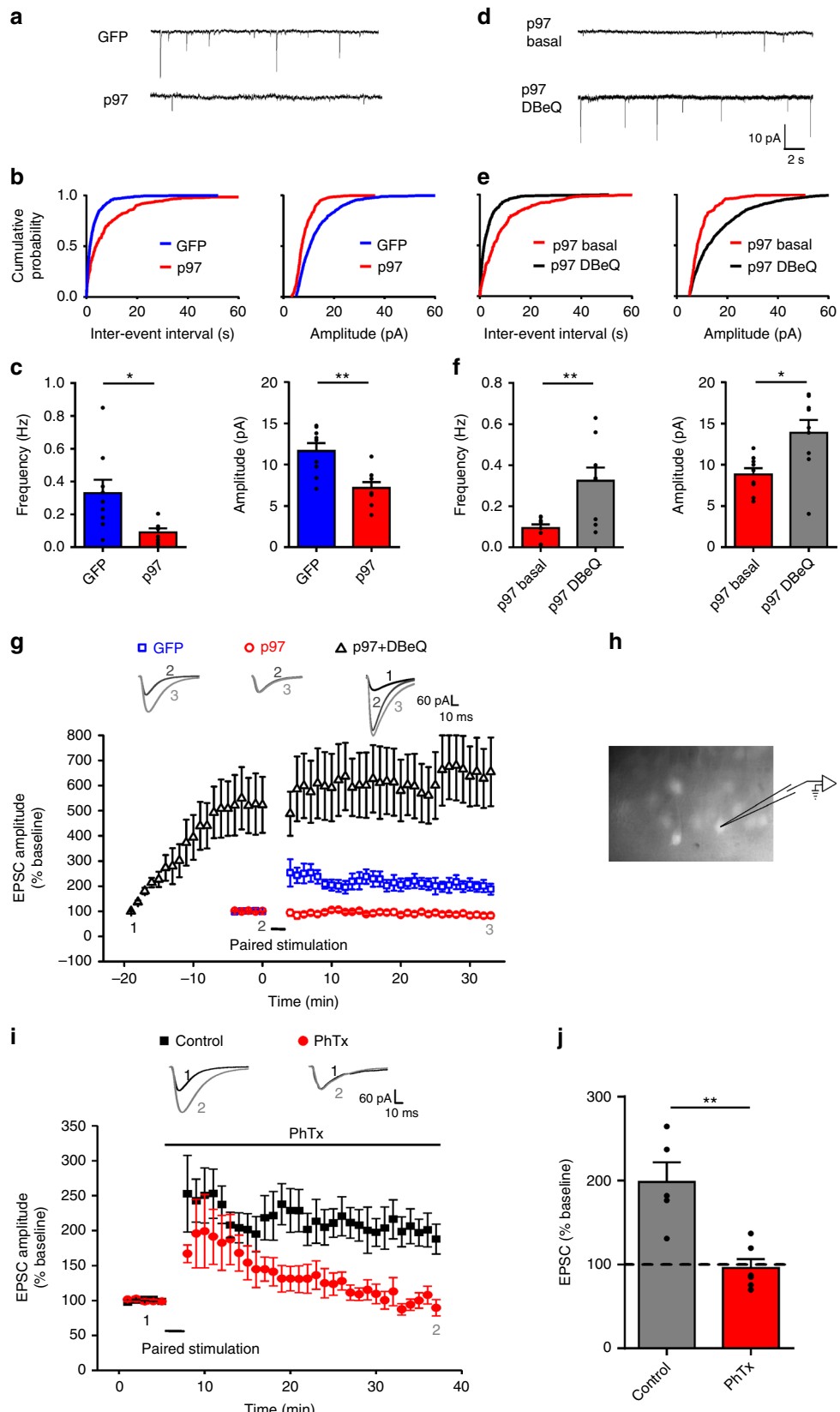

intracellular reserve pool under basal conditions. Furthermore, given that p97 also promotes the formation of GluA1-homo AMPARs at the expense of reducing the GluA1/GluA2 hetero-meric AMPARs, p97 appears to be an important molecular player in regulating the size of the intracellular reserve pool of GluA1-

homo AMPARs; thereby contributing significantly to the expression of various forms of GluA1-homo AMPAR-dependent synaptic and cellular plasticity.

Recent work by Plant et al. showed that there is a rapid and transient insertion of GluA1-homo AMPARs during the early

**Fig. 8** Overexpression of p97 promotes the formation and retention of GluA1-homo AMPARs and prevents LTP. AAV viral vector containing GFP or YFP-p2a-p97 was injected into the ventricle of mice at P0. mEPSCs (**a**–**f**) or evoked EPSCs (**g**, **i**) were recorded in visually identified GFP- or YFP-positive CA1 pyramidal neurons in hippocampal slices prepared at P14–21. Overexpression of p97 significantly decreased the amplitude and frequency of mEPSCs (**a**–**c**; $n = 9$ cells, *$p < 0.05$, **$p < 0.01$, two-tailed $t$ test), and this was reversed by including p97 inhibitor DBeQ in the recording pipette (**d**–**f**; $n = 9$ cells, *$p < 0.05$, **$p < 0.01$, two-tailed paired $t$ test). From **a** to **f** are represented traces (**a**, **d**), cumulative probability plots of inter-event intervals and amplitudes (**b**, **e**), and bar graphs of averaged frequencies and amplitudes of mEPSCs recorded in GFP or YFP-p2a-p97 (p97) expression cells (**c**), or in YFP-p2a-p97 expression cells with DBeQ (11 μM) added in the pipette solution 0–5 min (p97 basal) and 15–20 min (p97 DBeQ) after the formation of whole-cell configuration (**f**). **g** Overexpression of YFP-p2a-p97 (Red line with empty circles), but not GFP (blue line with empty squares), abolished paired-stimulation protocol to induce LTP at CA1 synapses. However, in neurons infected with p97, DBeQ (11 μM) contained in the recording pipette (black line with empty triangles) significantly increased baseline synaptic transmission, and occluded the further induction of LTP (GFP: $n = 5$ cells; p97: $n = 7$ cells; p97 + DBeQ: $n = 8$ cells). **h** Representative image shows GFP-positive neurons in CA1 region for recording. **i** In neurons expressing GFP, the paired stimulation reliably induced LTP only in control, nontreated neurons (black line with squares), but not in neurons treated with PhTx (10 μM, Red line with circles) applied just before the stimulation. **j** Bar graphs of averaged EPSCs recorded 25–30 min after paired stimulation show PhTx totally blocked LTP (Control: $n = 5$ cells; PhTx: $n = 6$ cells; **$p < 0.01$, two-tailed $t$ test). The error bars represent SEM

phase of hippocampal CA1 LTP[8]. Also, the activation of this subpopulation appears to be required for the full expression of LTP[8]. However, this work was challenged by a following study reported by Adesnik and Nicoll[9]. The reasons for the conflicting results of the studies remain to be determined. In our current study, we provided several lines of evidence supporting for the involvement of such a rapid insertion of GluA1-homo AMPARs in hippocampal LTP and a critical role of p97 in this process under our experimental conditions. First, using a well-studied chemical LTP model in cultured hippocampal neurons which share similar mechanisms with hippocampal slices[15,40], we showed a significant insertion of GluA1-homo AMPARs into synapses 5 min after LTP induction (Fig. 5a–e). Second, we found that this synaptic corporation of GluA1-homo AMPARs during LTP is accompanied by a rapid dissociation of p97 from the GluA1 complex (Fig. 5f, g). These results strongly suggest that disruption of the p97–GluA1 interaction may be a critical step in allowing GluA1-homo AMPARs to be released from the intracellular reserve pool and inserted into the plasma membrane during the early phase of LTP expression. Moreover, we demonstrated that inhibition of p97 activity by DBeQ, which can rapidly dissociate p97–GluA1 interaction, significantly increased the amplitude of the evoked EPSCs and occluded LTP production in hippocampal slices (Fig. 7b). This suggests that the dissociation of p97 from the GluA1 subunit may be sufficient to release GluA1-homo AMPARs from the intracellular reserve pool, thereby initiating their insertion into the plasma membrane. It is also interesting to note that overexpression of p97, while significantly reduced the frequency and amplitude of basal mEPSCs (Fig. 8a–c), could also prevent the production of LTP (Fig. 8g and Supplementary Fig. 1). This may imply that when p97 is expressed at an excessive level, it may result in its overwhelmingly strong association with GluA1 that cannot be sufficiently dissociated by the LTP-inducing signaling alone. Thus, it appears that both the activity and expression level of p97 can potentially alter its association with GluA1, and hence contribute to the regulation of LTP expression. Therefore, the identification of the molecules and/or signaling pathways downstream of NMDAR activation that modulate p97's activity, expression or its ability to interact with GluA1 should facilitate our understanding of p97's role in the molecular mechanisms of LTP.

In addition to the most well-characterized NMDAR-dependent LTP and LTD, activity-dependent modulation of GluA1-homo AMPARs on the neuronal surface have also been shown to be critically important in several other forms of synaptic and/or cellular plasticity[3–5]. These include homeostatic synaptic plasticity induced by chronically reducing neuronal activity[11,12], and a rapid increase of GluA1-homo AMPARs following global ischemic insults[13]. It would be very interesting to study whether

and how p97 is involved in mediating these forms of GluA1-homo AMPAR specific plasticity. Thus, the significance of our study may reach far beyond the NMDAR-dependent plasticity in the hippocampus.

## Methods

**Plasmids construction.** HA-GluA1, HA-GluA2, non-tagged GluA2, GluN1, and GluN2A were generated and published in out lab's previous publications[19,20]. GluK1 was a gift from Dr. Jim Huettner (Washington University School of Medicine in St. Louis). 6× His-p97 plasmid was a gift from Dr. Graham Warren (Yale University, School of Medicine). p97 expressed in mammalian cells was generated by PCR amplification of p97 and inserted into NotI and BamHI sites of pcDNA3.1 vector (Invitrogen). p97-GFP was generated by PCR amplification of p97 and inserted into KpnI and SmaI sites of pCAG-GFP vector (Addgene). pAAV-GFP under human synapsin promoter was a gift from Upenn Vector Core. A 2A peptide (GATNFSLLKQAGDVEENPGP) was used to link YFP with p97, and GFP was replaced by YFP-p2A-p97 to make pAAV-YFP-p2A-p97. GluA1-no-Ctail, GluA1C594, and GluA1C542 were generated using Quick-Change Site Directed Mutagenesis Kit (Stratagene) to mutate GluA1 809 GAG to TAG, 594 GGA to TGA, and 542 TTC to TGA, respectively. GluA1Δ510-541 was generated using Quick-Change Site Directed Mutagenesis Kit (Stratagene) to delete GluA1 510–541. To swap the N-terminal or C-terminal of GluA1 with GluA2 N-terminal or C-terminal, HindIII restriction sites were generated by Quick-Change Site Directed Mutagenesis Kit (Stratagene). After digestion by HindIII, the GluA2 N-terminal or C-terminal was ligated to the resulting backbone of HA-GluA1. GST-GluA1NT and GST-GluA2NT were generated by PCR amplification of GluA1NT (4–503) or GluA2NT (1–528) and inserted into EcoRI and SalI sites of pGEX 4T-1 vector (Amersham). All the constructions were confirmed by sequencing.

**Animals.** All experimental procedures with animals were conducted following the guidelines of the Canadian Council for Animal Care and approved by the University of British Columbia Animal Care Committee.

Hippocampal neuron cultures were prepared from embryonic day 18 rats. Electrophysiological recordings in Fig. 6g–k, Fig. 7, and Supplementary Fig. 2 were performed using Sprague–Dawley rats (2–4 weeks old). AAV injection was performed on C57BL/6 wild-type mice at postnatal day 0. Electrophysiological recordings in Fig. 8 and Supplementary Fig. 3 were performed using C57BL/6 mice (2–3 weeks old)

**Cell line culture and transfection.** COS7 (ATCC, Cat#CRL-1651) or HEK293 (ATCC, Cat#CRL-1573) cells were maintained in DMEM (Invitrogen) supplemented with 10% FBS. Cells were transfected using Lipofectamine 2000 (Invitrogen) following manufacture's instruction. Immunoprecipitation, immunofluorescence, or electrophysiological recording were performed 48 h after transfection.

**Immunoprecipitation and Western blotting.** Hippocampal slices, cultured hippocampal neurons or transfected COS7 cells were homogenized in ice-cold radioimmunoprecipitation assay (RIPA) buffer (50 mM Tris–HCl, pH 7.4, 1% Triton X-100, 150 mM NaCl, 1 mM EDTA, 0.5% deoxycholic acid sodium, and a cocktail of protease inhibitors (Roche Applied Science)). After incubation on ice for 30 min, the homogenates were centrifuged at $20,817 \times g$ at 4 ℃ for 15 min. The supernatants were collected and the total protein concentrations were measured using protein assay reagents (Bio-Rad Laboratories). For immunoprecipitation, 1 mg cell lysates were incubated with anti-GluA1 (lab raised rabbit polyclonal antibody against C-terminal 816–889), anti-GluA2 (lab raised rabbit polyclonal antibody against C-terminal 833–883), or anti-HA (Roche Applied Science,

Cat#11867431001) antibody in 1 mL of RIPA buffer for 4 h at 4 °C. Protein A-sepharose (GE Healthcare) was then added to the mixture and incubated for overnight. For immunoprecipitation of p97-GFP, GFP-Trap (Chromotek, Cat#gta-20) was incubated with cell lysates for 1 h at 4 °C. The complex was then isolated by centrifugation and washed twice with washing buffer (500 mM NaCl, 1% Triton X-100, 50 mM Tris–HCl, pH 7.4) and twice with phosphate-buffered saline (PBS). The precipitated proteins were eluted from the sepharose beads by boiling in 2× sample buffer at 60 °C for 5 min. Thirty microgram lysate was used as control for total protein expression level.

Proteins eluted from the beads or total lysates were subjected to a 10% sodium dodecyl sulfate polyacrylamide gel electrophoresis (SDS-PAGE) and were transferred to a polyvinylidene difluoride membrane. The membrane was blocked by 5% milk for 1 h at room temperature, immunoblotted, sequentially stripped, and reprobed on the same blot with anti-p97 (Fitzgerald, Cat#10R-P104A; 1:1000), anti-GluA1 (lab raised rabbit polyclonal antibody against C-terminal 816–889; 1:2500), anti-GluA2 (Millipore, Cat#MAB397; 1:1000), anti-HA (Roche Applied Science, Cat#11867431001; 1:1000), anti-GFP (Invitrogen, Cat#11122; 1:2000), anti-Na$^+$K$^+$ATPase (Abcam, Cat#ab7671; 1:2000), or anti-$\beta$-Actin (Sigma, Cat#AC74; 1:2500) antibody. Blots were developed using Amersham enhanced chemiluminescence detection kit (GE Healthcare) and imaged with the Bio-Rad gel imaging system (Bio-Rad Laboratories). Protein band intensities were quantified using Quantity One (Bio-Rad Laboratories). All the original blots images are provided as the Source Data file.

**Mass spectrometric analysis**. Protein bands of interest were excised from the SDS-PAGE gel and subjected to in-gel trypsinization. Digestions were carried out overnight at 37 °C with sequencing grade trypsin (Promega). Peptides were extracted, dried down and then reconstituted in 0.1% TFA and desalted with $\mu$-C18 ZipTips (Millipore). The peptides bound to the ZipTip were eluted out in 50:50 (v/v) ACN/0.1% Hac and mixed with a saturated solution of CHCA (in 30:70 v/v ACN/0.1% TFA) in the ratio of matrix to analyte of 1:1 (v/v). About 0.8 $\mu$L of the mixture was deposited onto a sample plate and air dried. Matrix-assisted laser desorption/ionization time-of-flight mass spectrometry (MS) was performed on 4700 Proteomics Analyzer (Applied Biosystems) using positive ion reflector mode. The peptide spectra were internally calibrated with bradykinin and ATCH peptides (Sigma) and processed with the 4700 Explorer software. The MS data was analyzed and searched for protein identification using the MASCOT search engine (http://www.matrixscience.com).

**GST pull down assay**. GST-GluA1NT, GST-GluA2NT and 6× his-p97 were expressed in BL21 *Escherichia coli* and purified from bacterial lysates according to the manufacturer's protocol (Pharmacia). Equal amount of glutathione sepharose conjugated GST-GluA1NT or GST-GluA2NT (adjusted by coomassie blue staining after resolved by SDS-PAGE) was incubated with 0.5 $\mu$g purified 6× his-p97 at 4 °C for overnight. After washing 4 times with PBS, the complexes were boiled in 2× sample buffer at 95 °C for 5 min. Proteins were resolved by SDS-PAGE, and probed with the corresponding antibodies as described in Section "Immunoprecipitation and Western Blotting".

**Biotinylation**. For biotinylation of surface proteins in COS7 cells, HEK293 cells, or cultured hippocampal neurons, cells were washed 3 times with ice-cold ECS (140 mM NaCl, 1.3 mM CaCl$_2$, 5.4 mM KCl, 1 mM MgCl, 25 mM HEPES, 33 mM glucose; pH 7.4), and incubated in the ECS containing 1 mg/ml EZ-link Sulfo-NHS-LC-Biotin (Thermo Fisher) at 4 °C for 30 min. Then the remaining active biotin was quenched by washing 3 times with ice-cold ECS containing 100 mM glycine. After homogenizing in RIPA buffer, the biotinylated proteins were isolated using streptavidin-conjugated sepharose beads (Sigma), eluted from the beads, resolved by SDS-PAGE, and immunoblotted with the corresponding antibodies as described in Section "Immunoprecipitation and Western Blotting".

**Neuron culture and transfection**. Medium-density hippocampal neurons were prepared from E18 rat embryos and maintained in Neurobasal medium (Invitrogen) containing B-27 supplement (Invitrogen). For Co-IP or biotinylation, dissociated hippocampal neurons were plated on 10 cm dishes coated with poly-L-lysine (PLL); and experiments were done on 14 days in vitro (DIV). For electrophysiology recordings, dissociated hippocampal neuron cultures were plated on glass coverslips coated with PLL and transfected with the plasmids using ProFection® Mammalian Transfection System (Promega) on 11 DIV; and electrophysiological recordings were performed on 13–14 DIV.

For immunostaining surface GluA1 or GluA2, cultured hippocampal neurons were prepared from embryonic day 18. The hippocampal cells were plated on glass coverslips coated with PLL at a density of 300,000 per 60 mm dish. The coverslips were inverted over a feeder layer of astrocytes in neurobasal medium (Invitrogen) supplemented with B27 (Invitrogen). Cytosine arabinoside (5 $\mu$M) was added to neuron culture dishes on 3 DIV to prevent overgrowth of glial cells. Neurons were transfected with YFP-p2a-p97 or YFP as control before plating using nucleofection (AMAXA Biosystems), and immunostaining was performed on 14 DIV.

**Immunofluorescent microscopy**. For detecting surface expression of GluA1 when overexpressing p97, COS7 cells were plated onto poly-D-lysine-coated glass coverslips set in 12-well culture dishes and transfected with 1 $\mu$g of the plasmid of interest. Forty-eight hours after transfection, surface GluA1 were live labeled with anti-HA antibody (Roche Applied Science, Cat#11867431001; 1:1000) for 30 min at 4 °C, followed by fixation with 4% paraformaldehyde in PBS for 10 min at room temperature. Surface GluA1 was visualized with an Alexa Fluor 555 anti-rat IgG antibody (Invitrogen; 1:500). Cells were then permeabilized using 0.2% Triton X-100 in PBS for 3 min. The internal GluA1 was labeled with anti-HA antibody (Roche Applied Science, Cat#11867431001; 1:1000), and visualized with an Alexa Fluor 488 goat anti-rat IgG antibody (Invitrogen; 1:500). Images were collected using LEICA DMIRE2 microscope. For quantification, the intensity of surface and internal GluA1 from randomly selected 7–10 cells of each coverslip were analyzed by Image J software to get an average ratio (surface vs. internal). The data from six to eight coverslips of each group were compared using two-tailed t-test.

For detecting surface GluA1 or GluA2 when overexpressing p97, cultured hippocampal neurons were live stained with anti-GluA1 (Calbiochem, Cat#PC246; 1:25) or GluA2 (Millipore, Cat#MAP397; 1:2000) antibody in conditioned media for 60 min at 37 °C. Neurons were then fixed, permeabilized, and incubated with anti-vGluT1 (Synaptic Systems, Cat#135304; 1:3000) and anti-MAP2 (Abcam, Cat#ab5392; 1:4000) antibodies, followed by the appropriate Alexa conjugated secondary antibody (1:500; Invitrogen). Images were acquired on a Zeiss Axioplan2 microscope with an oil immersion objective (63 × 1.4 numerical aperture) and Orca-Flash4.0 CMOS camera (Hamamatsu) using MetaMorph software (Molecular Devices) and customized filter sets. Synapses were identified as clusters with pixel overlap between the separately thresholded vGluT1 and GluA1 or GluA2 channels. The intensity of GluA1 or GluA2 on synapses was measured using Image J software.

**LTP and LTD induction in cultured hippocampal neurons**. For Co-IP, cultured hippocampal neurons were washed 3 times with ECS for 10 min each time, followed by application of glycine (200 $\mu$M; sucrose 100 mM; strychnine 5 $\mu$M; 3 min) or NMDA (10 $\mu$M; 3 min) in Mg$^{2+}$-free ECS to induce LTP or LTD, respectively. The cell lysates were collected at 10 min after stimulation unless indicated elsewhere. The p97–GluA1 complex was co-immunoprecipitated using anti-GluA1 antibody, and immunoblotted with the corresponding antibodies as described in Section "Immunoprecipitation and Western Blotting". For mEPSC recordings, cultured hippocampal neurons were perfused with ECS and recorded for baseline for 5 min, followed by application of glycine (200 $\mu$M; sucrose 100 mM; strychnine 5 $\mu$M; 3 min; in Mg$^{2+}$-free ECS) paired with depolarization to 0 mV. mEPSC was continuously recorded until 30 min after glycine application.

**AAV injection**. The packaging of AAV was performed by University of Pennsylvania Vector Core. On the day of birth (P0), C57BL/6 neonate mice were anesthetized using isoflurane and injected with 0.5 $\mu$l of viral vector into each cerebral lateral ventricle with a finely drawn glass pipette[41]. Electrophysiological recordings were performed on P14–21.

**Electrophysiological recordings**. For AMPAR-mediated currents in HEK293 cells, whole-cell recordings were performed at room temperature (20–22 °C) 48 h after transfection. The patch electrode solution contained the following (mM): Cs methane sulfonate, 130; EGTA, 0.5; Mg.ATP, 4; HEPES, 10; Na.GTP, 0.3; QX314. Br, 5; and NaCl, 8 (pH 7.25); and osmolarity between 280 and 290 mosmol$^{-1}$. The extracellular (perfusion or bathing) solution was of the following composition (mM): NaCl, 130; CaCl$_2$, 2; KCl, 2.5; MgCl$_2$, 2; HEPES, 10; glucose, 10; and sucrose, 10 (pH 7.4); and osmolarity between 300 and 310 mosmol$^{-1}$. Whole-cell currents were recorded at a holding potential of −60 mV unless indicated elsewhere. Rapid application/removal of receptor agonist was performed using a computer-controlled multi-barrel fast perfusion system (Warner Instruments). For recording rectification, spermine (100 $\mu$M) was added in the patch electrode solution. Transfected HEK 293 cells were held at −60, −40, −20, 0, +20, +40, +60 mV, and kainic acid (50 $\mu$M) was used to minimize receptor desensitization. For recording whole cell currents of GluA1, GluA2, GluK1, or GluN1/GluN2A, the saturating concentrations of agonist (1 mM glutamate, 10 mM glutamate, 1 mM glutamate, or 100 $\mu$M glutamate + 10 $\mu$M glycine, respectively) were used to maximize the difference in amplitude. The series resistance in these recordings varied between 6 and 8 M$\Omega$. Peak amplitude was analyzed using Clampfit 10.2 (Molecular Devices).

For recordings in the cultured hippocampal neurons, whole-cell recordings were performed at room temperature (20–22 °C) from the cultures 14–17 days after plating. The patch electrode solution contained the following (mM): Cs methane sulfonate, 130; EGTA, 0.5; Mg.ATP, 4; HEPES, 10; Na.GTP, 0.3; QX314.Br, 5; and NaCl, 8 (pH 7.25); and osmolarity between 280 and 290 mosmol$^{-1}$. The extracellular (perfusion or bathing) solution (ECS) was of the following composition (mM): NaCl, 140; CaCl$_2$, 1.3; KCl, 5.4; MgCl, 1; HEPES, 25; glucose, 33; TTX, 0.0005; and bicuculline methiodide, 0.02 (pH 7.4); and osmolarity between 310 and 320 mosmol$^{-1}$. Each cell was continuously perfused (1 ml/min) with this solution from a single barrel of a computer-controlled multi-barreled perfusion system. Solutions supplemented with glycine were applied from an alternative barrel. PhTx (Sigma, Cat#P207; 10 $\mu$M) was applied in the perfusion

solutions 5 min after the formation of whole cell configuration or glycine application. The series resistance in these recordings varied from 6 to 10 MΩ, and recordings in which series resistance varied by more than 10% were rejected. No electronic compensation for series resistance was employed. Cells that demonstrated a change in "leak" current of more than 10% were rejected from the analysis. For quantification of mEPSCs, the trigger level for detection of events was set approximately three times higher than the baseline noise. Inspection of the raw data was used to eliminate any false events. mEPSCs recorded in 3 min period of each condition were analyzed using Mini 6.0 software (Synaptosoft).

For recordings in hippocampal slices, hippocampal slices were prepared from Sprague–Dawley rats (2–4 weeks old) or AAV-injected C57BL/6 mice (2–3 weeks old) as previously described[19]. Briefly, rats or mice were anesthetized with 25% urethane. Brains were rapidly removed and were then cut into 400 μm thick coronal slices containing the hippocampus with a Leica vibratome (VT1200s, Leica) in ice cold artificial cerebrospinal fluid (ACSF) containing (mM) 125 NaCl, 2.5 KCl, 2 CaCl$_2$, 2 MgCl$_2$, 1.25 NaH$_2$PO$_4$, 26 NaCO$_3$, 25 glucose, osmolarity between 310 and 320 mosmol$^{-1}$ that was continuously bubbled with carbogen (95% O$_2$/5% CO$_2$) to adjust the pH to 7.35. Freshly cut slices were placed in a recovery chamber with carbogenated ACSF at 31 °C for 60 min, and then maintained at room temperature prior to recording. For whole-cell recording, slices were transferred to a recording chamber continuously perfused with carbogenated ACSF (2 ml/min) containing TTX (0.5 μM) and bicuculline methiodide (10 μM) for mEPSC recordings, or bicuculline methiodide (10 μM) for evoked EPSC recordings. All recordings were conducted at room temperature. Recording electrodes (3–6 MΩ) were constructed from borosilicate glass (1.5 mm OD/1.12 mm ID, World Precision Instruments) using a two-step horizontal puller (p-97, Sutter Instrument) and were filled with intracellular solution (pH 7.2 adjusted by CsOH, osmolarity between 290 and 300 mosmol$^{-1}$) containing (mM) 122.5 Cs-gluconate, 17.5 CsCl, 2 MgCl2, 10 HEPES, 0.2 EGTA, 4 ATP(K2) and, 5 QX314. EPSCs were evoked by stimulating the Schaffer collateral-commissural pathway via a constant current pulse (0.1 ms) delivered through a tungsten bipolar electrode. Synaptic responses were evoked at 0.05 Hz except during the LTP induction. CA1 pyramidal neurons were visualized using infrared differential interference contrast microscopy. For AAV-GFP and AAV-YFP-p2a-p97 mice, only GFP- or YFP-positive neurons were recorded. After formation of the whole cell configuration, the neurons were voltage clamped at −60 mV. Series resistances and input resistances were monitored. After obtaining a stable baseline for 5 min, paired low-frequency stimulations (120 pulses, 1 Hz) with postsynaptic depolarization to 0 mV for 3 min were used to induce LTP in patched neurons. To inhibit activity of p97, DBeQ (UBPBio, Cat#F1421; 11 μM) was added in the pipette solution. For AMPA/NMDA ratio recordings, AMPAR and NMDAR-mediated components of EPSCs were recorded under voltage clamped cell at −60 mV and +40 mV, respectively. For postsynaptic AMPAR rectification, spermine (100 μM) was added in the intracellular solution, and APV (100 μM) was added in the perfusion solution. For NMDAR mediated EPSC, whole-cell recordings were performed at holding potential of −30 mV in Mg$^{2+}$-free ACSF supplemented with DNQX (10 μM) to block AMPA receptors. Six to ten constitutive EPSCs were averaged.

All recordings were performed using a MultiClamp 700A amplifier (Molecular Devices), and records were filtered at 2 kHz, and acquired with pCLAMP 10 program (Molecular Devices).

**Statistical analysis**. Values are expressed as mean ± SME, and analyzed using a two-tailed student's $t$ test for comparison between two groups and ANOVA followed by post hoc (Fisher LSD) tests for comparisons among multiple groups. Statistical significance was defined as $p < 0.05$. No methods were used to determine whether the data met assumptions of the statistical approach. Each experiment was repeated at least in two independent experiments. Detailed information can be found in the figure legends.

**Reporting summary**. Further information on research design is available in the Nature Research Reporting Summary linked to this article.

## Data availability
All the original blots images are provided as the Source Data file. The data that support the findings of this study are available from the corresponding authors upon reasonable request.

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

## Acknowledgements

We thank Eva So for editorial assistance, Yuping Li for assisting with the neuron cultures, Dr. Ann Marie Craig (University of British Columbia) for C57BL/6 mice breeding for P0 AAV injection, and Dr. Graham Warren (Yale University, School of Medicine) for providing the p97 plasmid. This work was supported by Canadian Institutes of Health Research (Grant M0P-38090 and FDN-154286 to Y.T.W., Doctoral Fellowship to Y.G., Postdoctoral Fellowship to T.P.W.), Heart and Stroke Foundation of Canada (Focus on Stroke Doctor Research Awards to Y.G. and T.C.), UBC (University Graduate Fellowship to Y.G.), China Medical University Hospital (Postdoctoral Fellowship to Y.G.), the Taiwan Ministry of Science and Technology (MOST 107-2320-B-039-061-MY3 to D.W.), the Taiwan National Health Research Institutes (NHRI-EX108-10815NI to D.W.), and the China Medical University (CMU104-S-14-05 to D.W.).

## Author contributions

J.L. and Y.T.W. designed the screen and initiated the study. J.L. raised anti-GluA1 and GluA2 antibodies, performed initial co-IP experiment (Fig. 1b), and S.L. performed mass spectrometric analysis (Table 1). Y.G., M.T. and Y.T.W. designed the experiments beyond the mass spectrometric analysis. Y.G. performed plasmids construction, designed AAV vectors, and performed experiments and analyzed data for biochemical studies (Fig. 1a, c–m, Fig. 2e, f, Fig. 3a, Fig. 5f–i, and Fig. 6e, f), immunostaining (Fig. 2g, h and Fig. 3b), and electrophysiological recordings (Fig. 2a–d, Fig. 3d–f, Supplementary Fig. 1); M.T. performed and analyzed biochemical studies (Fig. 6a–d) and electrophysiological recordings (Fig. 6g–k, Fig. 7, Fig. 8, Supplementary Fig. 2, and Supplementary Fig. 3); L.L. performed and analyzed mEPSC recordings (Figs. 4 and 5a–e); T.P.W. performed mEPSC recordings (Supplementary Fig. 1); B.G. performed the preliminary experiments for overexpression of p97 in mice; D.W. performed electrophysiological recordings (Fig. 3c); T.C. provided technical instruction for cell surface immunostaining (Fig. 3b). J.K. and Y.T.W. supervised the experiments. Y.G. and Y.T.W. wrote the manuscript.

## Additional information

**Competing interests:** The authors declare no competing interests.

