## [Peer Review File · Nature Communications]

Reviewers' comments:

Reviewer #1 (Remarks to the Author):

In this study, the authors describe p97 as a protein that selectively interacts with GluA1 subunits and promotes the formation and sequestration of calcium permeable AMPA receptors away from synapses in hippocampal neurons. In this study, biochemical, electrophysiological and pharmacological evidence is provided from heterologous cells and dissociated and hippocampal slice preparations to support the idea that p97 supports the formation and sequestration of calcium permeable AMPA receptors (presumably GluA1 homomers) away from synapses under baseline conditions. The authors also suggest that LTP induction paradigms promote reductions in the ability of p97 to associate with GluA1 homomers causing GluA1 homomers to become synaptically expressed.

The question of the existence of calcium permeable AMPA receptors in hippocampal is an old and very controversial one. The topic tackled by the authors of this study is certainly interesting. Unfortunately, confidence in the conclusions drawn from present study is dramatically undermined by improperly controlled biochemical analyses and unsuitable execution of electrophysiological experiments. In its present state, it is not suitable for publication in any journal.

While there is a possibility that the present study is salvageable, a substantial amount of additional work must be performed. At this point the results are too preliminary to make a determination of whether the study is of significant impact to warrant publication in Nature Communications.

Major concerns:

- 1) In Figure 1, no input blots are provided for immunoprecipitation experiments. This is absolutely unacceptable and prevents any meaningful interpretation of the data.
- 2) In figure 2a, no quantification/statistics are provided for the data. Despite strong and important conclusions being drawn from the data, only sample traces are presented.
- 3) In Figure 2b-c AMPA receptor rectification data is presented. Despite the contents of the internal solutions being listed many times in the methods, spermine is never mentioned!! To resolve AMPAR rectification with whole cell patch clamping spermine must be in the internal solution. It would be very distressing if spermine was not included in the internal solution used to obtain Figure 2b.
- 4) In Figure 3e, demonstration that p97 does not inhibit GluA2 and/or GluA2/3 mediated currents is an important control to show specificity.
- 5) While it is unfortunate that the authors are unable to genetically manipulate p97 function and are forced to use a reported pharmacological inhibitor, this affords them the opportunity to perform potentially compelling experiments that are glaringly absent. The authors show that patching CA1 pyramidal neurons with DeBQ in the pipette produces a run up in AMPAR-eEPSC amplitude that is presumably due to synaptic insertion of Calcium permeable AMPARs. After currents have run up, the authors need to wash PhTX onto that same neuron. PhTX application must then rapidly reverse the run up produced by DBeQ in that cell. The authors must also show that inclusion of DeBQ in the patch pipette does not cause run up of NMDAR-eEPSCs. Furthermore, the authors must also show inclusion of DeBQ in the patch pipette produces a change in synaptic AMPAR rectification in neurons.
- 6) In Figure 6k, the authors conclude that LTP cannot be induced after waiting 20min subsequent to patching a cell using an internal solution that contains DeBQ. There is no way LTP can be induced after holding a whole cell patch for 20 min regardless of the internal solution. It is fairly

common knowledge in the field that an as yet unknown critical factor for LTP washes out of neurons under whole cell mode after the first 5 minutes of patching. The "LTP" shown in the control cell in figure 6k does convince this reviewer that this group has learned how to circumvent this well established limitation. If the authors must apply DBcQ via patch pipette rather than washing it onto neurons, this limitation precludes a direct investigation of endogenous p97's role in LTP.

7) In figure 7f, the authors show nice LTP induction in GFP expressing neurons after holding a cell for 5 min. p97 overexpression is shown to block LTP. The conclusion is that overexpression p97 may prevent GluA1 homomers from reaching synapses during LTP. While this may or may not be true, the authors seem to assume that insertion of Calcium permeable AMPA receptors during LTP support the AMPAR-potential seen during this process. What's puzzling to this reviewer is that the authors cite Adesnik and Nicoll, 2007 numerous times as support for this notion. In reality, Adesnik and Nicoll conclude precisely the opposite, providing very strong evidence against the insertion of calcium permeable AMPARs during slice LTP. Adesnik and Nicoll use the same slice preparations and a similar pairing LTP induction paradigm as the present study. Because of this, the authors must provide some evidence of their own that insertion of calcium permeable ampa receptors is occurring with their slice LTP. In other words, slice LTP shown in GFP transfected neurons in figure 7f must be shown to be affected by PhTx in some way. P97 overexpression must also be shown to not affect baseline NMDAR-eEPSCs in order to conclude that p97 overexpression impacts some mechanism downstream of NMDAR activation in LTP induction.

Reviewer #2 (Remarks to the Author):

In this study, Ge et al., identified p97, a type II AAA ATPase also called valonsin-containing protein (VCP), as a novel and unique GluA1 subunit-specific interacting protein. They found that p97 promotes the formation of the homomeric GluA1 AMPARs in the cytoplasmic compartment. This is important because it forms the reserve pools for AMPARs. They also found that p97 dissociated from the homomeric GluA1 AMPARs following the induction of LTP. Without the restrain of p97, the GluA1 homomeric AMPARs inserts into the postsynaptic membrane rapidly and results in the LTP. The experiments were well designed and the results look so clear and interesting. However, I have several major concerns as follows.

Fig 1,

The author should show their data for the antibody specific to GluA1 and GluA2 by western blot in hippocampal neurons or homogenates.

the author demonstrate that p97 only interacts with the GluA1 subunit of the homomeric GluA1, while they just performed co-IP using hippocampal homogenates with anti-GluA1 or anti-GluA2 antibody and immunoblotted the immunoprecipitates with anti-p97. They should perform the immunoprecipitation by anti-p97 and then immunoblot with anti-GluA1 and anti-GluA2 antibody to verify it in hippocampal homogenates or COS-7 cells with specific constructs transfection;

Fig. 3c, the author demonstrated overexpression of p97 significantly reduced the level of GluA1 on the cell surface, to consolidate their conclusion, the immunofluorescence with membrane receptor

immunostaining protocol should be employed to detect the membrane GluA1 and GluA2 level in the hippocampal neurons with or without the p97 overexpression.

Fig. 6, does the DBE-Q treatment disrupts the p97-GluA1 interaction at the dose-dependent manner?

Fig. 7, the virus injection into the mouse ventricles at P0 is very interesting, can the author provide the representative image for the virus infection in the slices?

Reviewer #3 (Remarks to the Author):

The MS by Ge et al. reports the identification of p97 as a novel potential GluA1 homomeric AMPAR interacting protein with the ability to regulate the formation and intracellular retention of GluA1 homomeric AMPARs.

Using mass spec analysis on GluA1- or GluA2-IP from brain lysates, the authors identified p97 as being associated with GluA1 but not GluA2-containing AMPARs. Using heterologous cells and transient over-expression of GluA1/GluA2/p97, the authors demonstrate that p97 associates with GluA1 homomeric AMPAR but not with GluA2-containing AMPAR. Moreover, the authors claim p97 as being important to the formation of GluA1 homomeric AMPARs in detriment of GluA2-containing AMPARs formation.

By using complementary approaches (IP, ICC, and electrophysiology in heterologous cells), the authors proceed by demonstrating that p97 association with GluA1 homomeric AMPAR not only promotes their formation but also regulates their surface expression by promoting their intracellular retention (fig2 and 3).

The authors then used the transient expression of GluA1 and p97 in dissociated primary neurons to verify that under basal condition, the over-expression of GluA1 alone resulted in significant surface expression of GluA1 homomeric AMPAR while co-expression with p97 prevented this, suggesting that p97 is important for the intracellular retention of GluA1 homomeric AMPAR. Moreover, the authors show that chemLTP, but not chemLTD, stimuli regulate the number of phalloidin sensitive AMPAR expression. This is associated, although disconnected at this point, from the observation that during chemLTP induction there is a time-dependent and transient decrease in p97-GluA1 association in the first 30min, with full recovery to the basal levels after 1h of chemLTP-induction.

In order to investigate the importance of p97, the authors used a p97 inhibitor, DBE-Q and overexpression of p97. DBE-Q reduced GluA1-p97 interaction and increased GluA1 surface expression in cultured neurons. Furthermore, in acute brain slices, intracellular application of DBE-Q resulted in an increase of mEPSC frequency, a gradual increase in the amplitude of evoked EPSCs and in AMPA/NMDA ratio. DBE-Q resulted in. Finally, using a pairing protocol to trigger LTP, the authors found that DBE-Q occluded LTP. On the other hand, p97-overexpression, fully blocked LTP.

Altogether, this MS reports an interesting finding, although falling short of demonstrating the mode of action of p97, specificity and the mechanism of the activity dependence of the putative p97-GluA1 interaction.

Major points:

- 1) The authors need to investigate the specificity of p97 action on a few related receptors such as KARs and NMDARs.
- 2) Also, regarding specificity, the authors need to establish the specificity of their antibodies in KO samples
- 3) Having some idea of the binding site between GluA1 and p97 would be interesting
- 4) There is no formal evidence at this point that the interaction between p97 and GluA1 is direct. This should be at least discussed.
- 4) Having some idea of the mechanism of GluA1-p97 dissociation during LTP would also be nice. For example is it CaMKII dependent?

5) Fig1g-h and Fig2d-e the effects are rather modest. Wouldn't the author's model suggest stronger effects ?

Minor points

- 1) Regarding the competition assay between GluA2 and p97, the authors need to show the blots that demonstrate that such reduction in GluA1-p97 interaction was not due to a decrease in their expression levels.
- 2) Fig5: why don't the authors study the effect of p97 overexpression and inhibition of chemLTP induced AMPAR modulation? what is the effect of manipulating P97 levels on the GluA1 dependent part of LTP?
- 3) Fig3e: modify to highlight the change in peak current
- 4) What is the difference between Fig 2d-e and fig 3ab ?

NCOMMS-18-20421

"Regulation of GluA1 homomeric AMPA receptor formation and plasma membrane expression by p97 (VCP)"

Response to Reviews

We thank all reviewers for their helpful comments and for their overall high level of enthusiasm for this work. We have now thoroughly addressed Reviewers' concerns with a series of new experiments and test revisions. The major new results presented in the revised manuscripts include 1) co-immunoprecipitation to show inhibition of p97 activity by DBeQ dose-dependently disrupts GluA1-p97 interaction; 2) the electrophysiological analysis to illustrate the run-up of eEPSCs by p97 inhibitor DBeQ is a result of p97-GluA1 disruption induced postsynaptic insertion of the GluA1 homomeric receptors; and 3) PhTx prevented LTP by blocking the GluA1 homomeric receptors under our experimental conditions. Additional experiments and further detailed point by point responses to reviewers are provided below. We thank the reviewers for their thoughtful and constructive comments and suggestions that, we believe, have significantly increased the significance and impact of our study.

Reviewer #1:

In this study, the authors describe p97 as a protein that selectively interacts with GluA1 subunits and promotes the formation and sequestration of calcium permeable AMPA receptors away from synapses in hippocampal neurons. In this study, biochemical, electrophysiological and pharmacological evidence is provided from heterologous cells and dissociated and hippocampal slice preparations to support the idea that p97 supports the formation and sequestration of calcium permeable AMPA receptors (presumably GluA1 homomers) away from synapses under baseline conditions. The authors also suggest that LTP induction paradigms promote reductions in the ability of p97 to associate with GluA1 homomers causing GluA1 homomers to become synaptically expressed.

The question of the existence of calcium permeable AMPA receptors in hippocampal is an old and very controversial one. The topic tackled by the authors of this study is certainly interesting. Unfortunately, confidence in the conclusions drawn from present study is dramatically undermined by improperly controlled biochemical analyses and unsuitable execution of electrophysiological experiments. In its present state, it is not suitable for publication in any journal.

While there is a possibility that the present study is salvageable, a substantial amount of additional work must be performed. At this point the results are too preliminary to make a determination of whether the study is of significant impact to warrant publication in Nature Communications.

We thank the reviewer for their valuable constructive suggestions. As detailed below, we have addressed these concerns and suggestions with a series of new experiments and revisions. We believe these have made our case much stronger.

Major concerns:

1) In Figure 1, no input blots are provided for immunoprecipitation experiments. This is absolutely unacceptable and prevents any meaningful interpretation of the data. *We apologize that the input blots were omitted and have added the input blots in Fig. 1.*

2) In figure 2a, no quantification/statistics are provided for the data. Despite strong and important conclusions being drawn from the data, only sample traces are presented.

Fig. 2a is the sample traces of AMPAR rectification recordings in HEK293 cells transiently expressing GluA1 alone, GluA2 alone, or GluA1/A2 with or without p97. The current amplitudes were measured and I-V relationship curve was shown in Fig. 2b, and quantification of rectification index ($I_{+60/-60}$) was shown as bar graphs in Fig. 2c. We apologize that we didn't include the quantification to compare whole-cell currents of GluA1/A2 when overexpressing p97 and draw the conclusion of "More pronouncedly, addition of p97 also drastically reduced the current amplitude of whole-cell currents (Fig. 2a; A1/A2+p97) in comparison with currents from cells only expressing GluA1 and GluA2 (Fig. 2a; A1/A2+Vec)." We have now added this quantification in Fig. 2d to show overexpression of p97 decreases GluA1/A2 whole-cell currents.

3) In Figure 2b-c AMPA receptor rectification data is presented. Despite the contents of the internal solutions being listed many times in the methods, spermine is never mentioned!! To resolve AMPAR rectification with whole cell patch clamping spermine must be in the internal solution. It would be very distressing if spermine was not included in the internal solution used to obtain Figure 2b.

We apologize that we forgot to mention that spermine (100 μ M) was added in the intracellular solution in the original manuscript and thank the reviewer for pointing out this mistake. We have now added it in the method of the revised manuscript.

4) In Figure 3e, demonstration that p97 does not inhibit GluA2 and/or GluA2/3 mediated currents is an important control to show specificity.

We thank the reviewer for this excellent suggestion (this was also mentioned by Reviewer 3). We fully agree with the reviewers that testing effects of p97 on the function of AMPARs with other glutamate receptor subunit compositions, such as GluA2 homomeric AMPARs, Kainate receptors or NMDA receptors, is a good control. Therefore, we performed whole-cell current recordings in HEK293 cells expressing GluA2, GluK1, or GluN1/N2A with or without co-transfection of p97. As shown in Fig. 3d-f, overexpression of p97 doesn't significantly alter the currents mediated by either GluA2 or GluK1 or GluN1/N2A receptors.

5) While it is unfortunate that the authors are unable to genetically manipulate p97 function and are forced to use a reported pharmacological inhibitor, this affords them the opportunity to perform potentially compelling experiments that are glaringly absent. The authors show that patching CA1 pyramidal neurons with DeBQ in the pipette produces a run up in AMPAR-eEPSC amplitude that is presumably due to synaptic insertion of Calcium permeable AMPARs. After currents have run up, the authors need to wash PhTx onto that same neuron. PhTx application must then rapidly reverse the run up produced by DBeQ in that cell. The authors must also show that inclusion of DeBQ in the patch pipette does not cause run up of NMDAR-eEPSCs. Furthermore, the authors must also show inclusion of DeBQ in the patch pipette produces a change in synaptic AMPAR rectification in neurons.

These are all excellent points, and in line with the reviewer's suggestions, we have now added new data to show when PhTx was bath applied immediately after the formation of whole-cell configuration, the run-up of eEPSC was totally blocked (Fig. 7f). Moreover, if PhTx was bath applied after eEPSC reached plateau, the run-up of eEPSC was reversed back to baseline level after 20 min PhTx application (Fig. 7g and h). These results suggest that inhibition of p97

activity caused the postsynaptic insertion of the GluA1 homomeric receptors. To further confirm this, we performed experiments to test AMPAR rectification by recording AMPAR mediated eEPSCs at various holding membrane potentials. Inclusion of DBeQ in the recording pipette shifted the I-V curve to an inward rectification (Fig 7c-e). To show the specificity, we recorded NMDAR mediated eEPSC in CA1 neurons, and inclusion of DBeQ in the pipette solution didn't change NMDAR mediated eEPSCs (Fig. S2).

6) In Figure 6k, the authors conclude that LTP cannot be induced after waiting 20min subsequent to patching a cell using an internal solution that contains DeBQ. There is no way LTP can be induced after holding a whole cell patch for 20 min regardless of the internal solution. It is fairly common knowledge in the field that an as yet unknown critical factor for LTP washes out of neurons under whole cell mode after the first 5 minutes of patching. The "LTP" shown in the control cell in figure 6k does convince this reviewer that this group has learned how to circumvent this well established limitation. If the authors must apply DBeQ via patch pipette rather than washing it onto neurons, this limitation precludes a direct investigation of endogenous p97's role in LTP.

To minimize potential complications from presynaptic effects, we applied DBeQ through the recording pipette instead of bath application. We agree with the reviewer that there are several previous studies reporting that it is very difficult to induce LTP 10 minutes after the formation of whole-cell recording due to the potentially washing-out effects. However, there are also some other studies that showed robust LTP induction after 15-40 min of whole-cell baseline recordings¹⁻⁴. Therefore, it appears whether LTP can be induced 10 min after initiation of whole-cell recordings may be very dependent on the different experimental conditions in each lab. As shown in Fig. 7b, under our experimental conditions, we could stably induce LTP after a 20 min of whole-cell baseline recordings in control cells. We think that our successful rates may have been increased by using a high resistance (6-7 MΩ) pipette with narrow tip opening and using a higher ZAP voltage, instead of using suction, to break the cell membrane. We found both can help to slow down the washout process.

7) In figure 7f, the authors show nice LTP induction in GFP expressing neurons after holding a cell for 5 min. p97 overexpression is shown to block LTP. The conclusion is that overexpression p97 may prevent GluA1 homomers from reaching synapses during LTP. While this may or may not be true, the authors seem to assume that insertion of Calcium permeable AMPA receptors during LTP support the AMPAR-potentialiation seen during this process. What's puzzling to this reviewer is that the authors cite Adesnik and Nicoll, 2007 numerous times as support for this notion. In reality, Adesnik and Nicoll conclude precisely the opposite, providing very strong evidence against the insertion of calcium permeable AMPARs during slice LTP. Adesnik and Nicoll use the same slice preparations and a similar pairing LTP induction paradigm as the present study. Because of this, the authors must provide some evidence of their own that insertion of calcium permeable ampa receptors is occurring with their slice LTP. In other words, slice LTP shown in GFP transfected neurons in figure 7f must be shown to be affected by PhTx in some way. P97 overexpression must also be shown to not affect baseline NMDAR-eEPSCs in order to conclude that p97 overexpression impacts some mechanism downstream of NMDAR activation in LTP induction.

First of all, we are very sorry for the rather confusing citations in our original manuscript, and we should have done a much better job in describing the controversies of the GluA1 homomeric AMPARs in LTP induction and expression. The reviewer is absolutely right that Adesnik and Nicoll's study is opposite to that of Plant et al 2006 study, providing strong evidence for the absence of calcium-permeable AMPARs during LTP induction and expression. However, both

studies are in a good agreement on the lack of calcium-permeable AMPARs under basal conditions. We had intended to cite the two studies as supporting evidence for the absence of the GluA1 homomeric receptors on the plasma membrane surface under basal conditions only, but mistakenly also cited both for the GluA1 homomeric receptor insertion during early phase LTP. This has been corrected in the revised manuscript. Regarding the discrepancy between our present study (that showed important role of the GluA1 homomeric receptor insertion at the early phase LTP) and Adesnik and Nicoll's study (that demonstrated no insertion of calcium-permeable AMPARs), we have no good explanation yet. However, we did observe several differences in experimental conditions between the two studies. For instances, in the study of Adesnik and Nicoll, slices were placed at room temperature for 1-2 hours during the recovery period; in the present study, slices were incubated at 31 °C for at least 60 min before recording. Also, the induction protocols were different: while Adesnik and Nicoll used a stimulation protocol of 2 Hz paired with a depolarization of neuron to 0-10 mV for 60 s, we used the stimulation protocol of 120 pulses at 1 Hz paired with a membrane depolarization of the neuron to 0mV for 3 min. Following suggestions of the reviewer, and to further support the involvement of the GluA1 homomeric receptor insertion during the expression of early LTP under our experimental conditions, we performed additional experiments showing that bath application of PhTx (10 μ M) blocked the ability of paired stimulation to induce LTP (Fig. S3) and that overexpression of p97 didn't change NMDAR mediated eEPSCs (Fig. S4).

Reviewer #2:

In this study, Ge et al., identified p97, a type II AAA ATPase also called valonsin-containing protein (VCP), as a novel and unique GluA1 subunit-specific interacting protein. They found that p97 promotes the formation of the homomeric GluA1 AMPARs in the cytoplasmic compartment. This is important because it forms the reserve pools for AMPARs. They also found that p97 dissociated from the homomeric GluA1 AMPARs following the induction of LTP. Without the restrain of p97, the GluA1 homomeric AMPARs inserts into the postsynaptic membrane rapidly and results in the LTP. The experiments were well designed and the results look so clear and interesting. However, I have several major concerns as follows.

We thank the reviewer for their encouraging evaluations on our study and we are also very grateful for constructive suggestions which, as detailed bellow, we have acted upon in our revision.

Fig 1, The author should show their data for the antibody specific to GluA1 and GluA2 by western blot in hippocampal neurons or homogenates.

We fully agree with the reviewer that it is critical to demonstrate the specificity of the antibodies in neuronal tissues we used in the present study. Due to the similar molecular weight of GluA1 and GluA2, it is hard to differentiate them from each other using Western Blot in hippocampal neurons. Another useful tool to test antibody specificity is to use GluA1 or GluA2 KO samples which was also suggested by Reviewer 3. However, GluA1 or GluA2 KO mice are beyond the reach of our lab. We looked into the possibility of ordering from the Jackson Laboratory (stock No: 019011 and 003143), but were told that the orders would need cryorecovery and long shipping/waiting time. If we also consider the breeding time for homozygous KO, this will take too long to obtain the KO samples. So instead of using KO samples as negative controls, we used GluA1 or GluA2 non-expressing cells (COS7 cells) transfected with GluA1 or GluA2 as positive controls. As shown in Fig. 1a, anti-GluA1 or anti-GluA2 can specifically immunoprecipitate HA-GluA1 or HA-GluA2 from transfected COS7 cells. In addition, we have validated the interaction of GluA1-p97 by co-immunoprecipitation of HA-GluA1 and p97 using

anti-HA antibody (Fig. 1e), or co-immunoprecipitation of p97-GFP and HA-GluA1 using GFP-Trap (Fig. 1f) in co-transfected COS7 cells.

the author demonstrate that p97 only interacts with the GluA1 subunit of the homomeric GluA1, while they just performed co-IP using hippocampal homogenates with anti-GluA1 or anti-GluA2 antibody and immunoblotted the immunoprecipitates with anti-p97. They should perform the immunoprecipitation by anti-p97 and then immunoblot with anti-GluA1 and anti-GluA2 antibody to verify it in hippocampal homogenates or COS-7 cells with specific constructs transfection; *We thank the reviewer for the suggested experiments. We have performed an additional set of experiments as suggested by the reviewer, and showed the specific co-immunoprecipitation of GluA1, but not GluA2, with p97 using GFP-Trap (Chromotek) from COS7 cells co-transfected with p97-GFP and HA-GluA1 or HA-GluA2 (Fig. 1f).*

Fig. 3c, the author demonstrated overexpression of p97 significantly reduced the level of GluA1 on the cell surface, to consolidate their conclusion, the immunofluorescence with membrane receptor immunostaining protocol should be employed to detect the membrane GluA1 and GluA2 level in the hippocampal neurons with or without the p97 overexpression.

In line with the suggestion of the reviewer, we performed immunofluorescent staining of surface GluA1 and GluA2 subunits in cultured hippocampal neurons overexpressing YFP-p2a-p97 or GFP and showed that overexpression of p97 decreased postsynaptic surface level of GluA1 without affecting that of GluA2 (Fig. 2f and g).

Fig. 6, does the DBeQ treatment disrupts the p97-GluA1 interaction at the dose-dependent manner?

We thank the reviewer for the excellent suggestion. In line with the suggestion, we performed DBeQ treatment at various doses (0, 0.1, 1, 10, 50 μ M) in COS7 cells that are transiently expressing p97 and HA-GluA1. As shown in Fig. 6e and f, DBeQ dissociates p97-GluA1 interaction in a dose-dependent manner, with an IC_{50} of 1.09 μ M. And at the concentration (11 μ M) we used for the electrophysiology experiments, DBeQ inhibits p97-GluA1 interaction at a nearly maximal level.

Fig. 7, the virus injection into the mouse ventricles at P0 is very interesting, can the author provide the representative image for the virus infection in the slices?

Following the request, we have now added the representative image in Fig. 8h.

Reviewer #3:

The MS by Ge et al. reports the identification of p97 as a novel potential GluA1 homomeric AMPAR interacting protein with the ability to regulate the formation and intracellular retention of GluA1 homomeric AMPARs.

Using mass spec analysis on GluA1- or GluA2-IP from brain lysates, the authors identified p97 as being associated with GluA1 but not GluA2-containing AMPARs. Using heterologous cells and transient over-expression of GluA1/GluA2/p97, the authors demonstrate that p97 associates with GluA1 homomeric AMPAR but not with GluA2-containing AMPAR. Moreover, the authors claim p97 as being important to the formation of GluA1 homomeric AMPARs in detriment of GluA2-containing AMPARs formation.

By using complementary approaches (IP, ICC, and electrophysiology in heterologous cells), the authors proceed by demonstrating that p97 association with GluA1 homomeric AMPAR not only promotes their formation but also regulates their surface expression by promoting their intracellular retention (fig2 and 3).

The authors then used the transient expression of GluA1 and p97 in dissociated primary neurons to verify that under basal condition, the over-expression of GluA1 alone resulted in significant surface expression of GluA1 homomeric AMPAR while co-expression with p97 prevented this, suggesting that p97 is important for the intracellular retention of GluA1 homomeric AMPAR. Moreover, the authors show that chemLTP, but not chemLTD, stimuli regulate the number of phalantotoxin sensitive AMPAR expression. This is associated, although disconnected at this point, from the observation that during chemLTP induction there is a time-dependent and transient decrease in p97-GluA1 association in the first 30min, with full recovery to the basal levels after 1h of chemLTP-induction.

In order to investigate the importance of p97, the authors used a p97 inhibitor, DBeQ and overexpression of p97. DBeQ reduced GluA1-p97 interaction and increased GluA1 surface expression in cultured neurons. Furthermore, in acute brain slices, intracellular application of DBeQ resulted in an increase of mEPSC frequency, a gradual increase in the amplitude of evoked EPSCs and in AMPA/NMDA ratio. DBeQ resulted in. Finally, using a pairing protocol to trigger LTP, the authors founds that DBeQ occluded LTP. On the other hand, p97-overexpression, fully bloqued LTP.

Altogether, this MS reports an interesting finding, although falling short of demonstrating the mode of action of p97, specificity and the mechanism of the activity dependence of the putative p97-GluA1 interaction.

We thank this reviewer for all the constructive suggestions. Below is the summary of the new experiments and edits that we have done to address the reviewer's concerns and suggestions.

Major points:

1) The authors need to investigate the specificity of p97 action on a few related receptors such as KARs and NMDARs.

We thank the reviewer for this excellent suggestion, a point also raised by Reviewer 1. As we described in our response to the reviewer 1, we performed whole-cell current recordings in HEK293 cells expressing GluA2, GluK1, or GluN1/N2A. As shown in Fig. 3d-f, overexpression of p97 didn't significantly change the current levels gated through either receptors.

2) Also, regarding specificity, the authors need to establish the specificity of their antibodies in KO samples

This is another important point, and also raised by Reviewer 2. We agree with both reviewers that the KO samples are very useful tools to test antibody specificity. However, as we mentioned in our response to the reviewer 2 above, GluA1 or GluA2 KO mice are currently beyond the reach of our lab. We looked into the possibility of ordering from the Jackson Laboratory (stock No: 019011 and 003143), but were told that the orders would need cryorecovery and at least an extra 11-14 weeks' shipping time. It is also relevant to point out that Jackson Laboratory cannot guarantee the success of cryorecovery, and may need a second round of attempt. Also, once we receive the heterozygous knockout, it will also take a very long time to breed to get the homozygous GluA1 or GluA2 knockout mice. So for both economic and time efficiency, instead of using KO samples as negative controls, we used GluA1 or GluA2 non-expressing cells (COS7 cells) transfected with GluA1 or GluA2 subunits as positive controls. As shown in Fig. 1a, anti-GluA1 or anti-GluA2 can specifically immunoprecipitate HA-GluA1 or HA-GluA2 from transfected COS7 cells. In addition, we have also validated the interaction of GluA1-p97 by co-immunoprecipitation of HA-GluA1 and p97 using anti-HA antibody (Fig. 1e), or co-immunoprecipitation of p97-GFP and HA-GluA1 using GFP-Trap (Fig. 1f) in co-transfected COS7 cells.

3) Having some idea of the binding site between GluA1 and p97 would be interesting

We thank the reviewer's suggestion. To map the binding domain of GluA1, we have designed several deletion or swap versions of mutant GluA1 constructs and performed co-immunoprecipitation in HEK cells transiently transfected with these constructs along with p97. As shown in Fig. 1h, deletion of TM1, M2, TM3, S2, TM4 or C-terminal of GluA1 didn't affect the p97-GluA1 interaction. However, swapping GluA1 N-terminal with GluA2 N-terminal completely abolished the p97-GluA1 interaction, suggesting the interaction domain is in the N-terminal of GluA1.

4) There is no formal evidence at this point that the interaction between p97 and GluA1 is direct. This should be at least discussed.

To test whether the p97-GluA1 interaction is a direct protein-protein binding, we performed in-vitro recombinant protein binding (GST pull down) assay using purified GST-GluA1NT or GST-GluA2NT and 6×his tagged p97. As shown in Fig. 1i, p97 was only pulled down by GST-GluA1NT, but not GST-GluA2NT, suggesting the p97-GluA1 interaction is direct and the interaction domain is in the N-terminal of GluA1.

4) Having some idea of the mechanism of GluA1-p97 dissociation during LTP would also be nice. For example is it CaMKII dependent?

p97 is an ATPase and its function is largely dependent on its ATPase activity. Therefore, in line with the suggestion of the reviewer, we first focused on the role of its ATPase activity in the p97-GluA1 association/dissociation under basal conditions and during LTP production using the p97 inhibitor DBEQ. We found that inhibition of p97 activity dose-dependently disrupted p97-GluA1 interaction, and thereby promoted surface insertion of the GluA1 homomeric AMPARs (Fig. 6a-f). As a result, DBEQ causes an increase in basal EPSCs and thereby occludes the LTP production. So the p97-GluA1 dissociation during LTP may be mainly dependent on the activity of p97. It was reported that activity of p97 is regulated by post-translational modifications, such as phosphorylation and acetylation⁵. 66 phosphorylation sites and 24 acetylation sites on p97 have been identified^{6,7}. However, the relevant enzymes and functional consequences of these modifications are poorly understood. Future study may need to identify detailed molecules and/or the signaling pathways downstream of NMDAR activation which directly modulate the activity of p97, and hence p97-GluA1 dissociation during LTP.

5) Fig1g-h and Fig2d-e the effects are rather modest. Wouldn't the author's model suggest stronger effects ?

The Fig 1l-m (previous Fig 1g-h) and Fig 2e-f (previous Fig 2d-e) are from the cells transfected with GluA1 and GluA2, which form mixed populations of GluA1/A2 heteromeric, GluA1 and GluA2 homomeric receptors. Although we do not know the exact proportion of each of these populations expressed in these cells, the rather modest effects of p97 may suggest that under such co-expression conditions, the overexpressed recombinant AMPARs may be primarily GluA1/GluA2 heteromeric, with very small proportions of them being GluA1 or GluA2 homomeric receptors.

Minor points

1) Regarding the competition assay between GluA2 and p97, the authors need to show the blots that demonstrate that such reduction in GluA1-p97 interaction was not due to a decrease in their expression levels.

We have now added the input blots in Fig. 1j to show the expression levels were not changed.

2) Fig5: why don't the authors study the effect of p97 overexpression and inhibition of chemLTP induced AMPAR modulation? what is the effect of manipulating P97 levels on the GluA1 dependent part of LTP?

It is a good idea to study the effect of p97 overexpression in glycine induced chemical LTP. We now show that overexpression of p97 in cultured hippocampal neurons blocked chemical LTP (Fig. S1a-e). Consistent with our findings in hippocampal brain slices (Fig. 8g), we reasoned that this is due to the overwhelming amount of p97 following overexpression promoting a strong association between p97-GluA1. This association is strong enough to prevent the LTP signaling-induced dissociation of GluA1 from p97, thereby preventing glycine-induced surface insertion of the GluA1 homomeric receptors and hence LTP expression.

3) Fig3e: modify to highlight the change in peak current

We have modified the representative traces in Fig. 3c (previous Fig. 3e) to show the change in peak current.

4) What is the difference between Fig 2d-e and fig 3ab?

We are sorry for not describing the results properly. Fig. 2e-f (previously Fig. 2d-e) showed result of the biotinylation experiments that reveal differential effects of p97 overexpression on the cell surface expression levels of GluA1 and GluA2 from HEK293 cells co-transfected with GluA1 and GluA2 subunits. As we mentioned before, these transiently expressed GluA1 and GluA2 form mixed population of the GluA1/A2 heteromeric receptors as well as GluA1 or GluA2 homomeric receptors. Fig. 2e-f (previously Fig. 2d-e) show that overexpression of p97 specifically decreased surface level of GluA1, explaining the outward shift of I-V curve as we observed in Fig. 2a-c. Because p97 only interacts with the GluA1 homomeric receptors, the decreased surface level of GluA1 is most likely due to the specific p97-GluA1 interaction-induced intracellular retention of the GluA1 homomeric receptors. Fig. 3a (previously Fig. 3a and b) directly tested the effects of p97 in retaining GluA1 in cells co-overexpressing p97 and GluA1 alone (without GluA2). In these experiments, we were able to show that overexpression of p97 indeed decreased surface trafficking of the GluA1 homomeric receptors.

References

1. Kato, K., Clifford, D.B. & Zorumski, C.F. Long-term potentiation during whole-cell recording in rat hippocampal slices. *Neuroscience* **53**, 39-47 (1993).
2. Duffy, S.N. & Nguyen, P.V. Postsynaptic application of a peptide inhibitor of cAMP-dependent protein kinase blocks expression of long-lasting synaptic potentiation in hippocampal neurons. *J Neurosci* **23**, 1142-50 (2003).
3. Kim, C.H. & Lisman, J.E. A role of actin filament in synaptic transmission and long-term potentiation. *J Neurosci* **19**, 4314-24 (1999).
4. Otmakhova, N.A., Otmakhov, N., Mortenson, L.H. & Lisman, J.E. Inhibition of the cAMP pathway decreases early long-term potentiation at CA1 hippocampal synapses. *J Neurosci* **20**, 4446-51 (2000).
5. Mori-Konya, C. et al. p97/valosin-containing protein (VCP) is highly modulated by phosphorylation and acetylation. *Genes Cells* **14**, 483-97 (2009).
6. Hornbeck, P.V. et al. PhosphoSitePlus, 2014: mutations, PTMs and recalibrations. *Nucleic Acids Res* **43**, D512-20 (2015).
7. Hanzelmann, P. & Schindelin, H. The Interplay of Cofactor Interactions and Post-translational Modifications in the Regulation of the AAA+ ATPase p97. *Front Mol Biosci* **4**, 21 (2017).

REVIEWERS' COMMENTS:

Reviewer #1 (Remarks to the Author):

In the revised manuscript the authors have done a nice job of addressing the concerns of all reviewers. My final request is that the authors include Figure S3 in a main figure and provide detailed product information in the methods regarding the philanthotoxin-433 used in this and other experiments in the study. I now recommend publication on the condition that these changes are made.

Reviewer #2 (Remarks to the Author):

The author fully addressed my queries. I have no further questions.

NCOMMS-18-20421B

"p97 regulates GluA1 homomeric AMPA receptor formation and plasma membrane expression "

Response to Reviews

We thank all reviewers for their helpful comments and for their overall high level of enthusiasm for this work. Detailed point by point responses to reviewers are provided below. .

Reviewer #1 (Remarks to the Author):

In the revised manuscript the authors have done a nice job of addressing the concerns of all reviewers. My final request is that the authors include Figure S3 in a main figure and provide detailed product information in the methods regarding the philanthotoxin-433 used in this and other experiments in the study. I now recommend publication on the condition that these changes are made.

We thank the reviewer for this suggestion and now included previous Supplementary Fig. 3 to the main Fig. 8 i and j. We have also added the detailed product information of philanthotoxin-433 in the Methods.

Reviewer #2 (Remarks to the Author):

The author fully addressed my queries. I have no further questions.

We thank this reviewer.